# FINITE-TIME CONVERGENCE ANALYSIS OF ACTOR-CRITIC WITH EVOLVING REWARD

## ABSTRACT

Many popular practical reinforcement learning (RL) algorithms employ evolving reward functions—through techniques such as reward shaping, entropy regularization, or curriculum learning—yet their theoretical foundations remain underdeveloped. This paper provides the first finite-time convergence analysis of a single-timescale actor-critic algorithm in the presence of an evolving reward function under Markovian sampling. We consider a setting where the reward parameters may change at each time step, affecting both policy optimization and value estimation. Under standard assumptions, we derive non-asymptotic bounds for both actor and critic errors. Our result shows that an $O(1/\sqrt{T})$ convergence rate is achievable, matching the best-known rate for static rewards, provided the reward parameters evolve slowly enough. This rate is preserved when the reward is updated via a gradient-based rule with bounded gradient and on the same timescale as the actor and critic, offering a theoretical foundation for many popular RL techniques. As a secondary contribution, we introduce a novel analysis of distribution mismatch under Markovian sampling, improving the best-known rate by a factor of $\log^2 T$ in the static-reward case.

## 1 INTRODUCTION

Reinforcement Learning (RL, Sutton et al. 1998) has attracted great research interest in the past decades. On the empirical side, a variety of practical algorithms have been proposed and have demonstrated remarkable success in a wide range of real-world scenarios (Mnih et al., 2013; Lillicrap et al., 2016; Ouyang et al., 2022; DeepSeek-AI et al., 2025). On the theoretical side, great efforts have been made to bridge the gap between empirical practice and theoretical foundations, providing rigorous convergence guarantees and solid theoretical understandings to the empirically powerful algorithms (Agarwal et al., 2021; Mei et al., 2020; Kumar et al., 2023; Wu et al., 2020; Olshevsky & Gharesifard, 2023).

Still, a very common setting adopted by many practical RL algorithms has been largely overlooked by theoretical analyses. RL theory is built upon Markov Decision Processes (MDPs), which typically assume the existence of a static underlying reward function, and the goal is to learn a policy that maximizes the expected cumulative reward. However, when applying RL in many real-world scenarios, a significant impediment is the difficulty of designing a reward function that is both learnable and aligns with the desired task. This challenge has led to the development of techniques that utilize evolving rewards. These include:

- **Reward Shaping:** Adding auxiliary rewards to guide the policy to the desired goal (Ng et al., 1999; Pathak et al., 2017; Burda et al., 2019; Zheng et al., 2018; Hu et al., 2020; Mahankali et al., 2024; Ma et al., 2024). The auxiliary rewards can come from prior knowledge, or be learned in a self-supervised manner during the training process.

- **Entropy or KL Regularization:** Introducing an entropy or KL regularization term to the optimization objective, which is equivalent to modifying the reward according to the current policy (Haarnoja et al., 2017; 2018a;b; Jaques et al., 2019; Stiennon et al., 2020). The regularization factor can be automatically adjusted during training.

- **Curriculum Learning:** Starting with easier tasks (and their associated rewards) and gradually increasing the difficulty (Narvekar et al., 2020).

Intuitively, a slight change of the reward function does not drastically alter the solution of the underlying MDP, allowing an RL algorithm to remain effective. However, when this change is negligible compared to the under-training policy or value function, the algorithm's effectiveness becomes questionable, as they are closely interconnected. Therefore, to rigorously support the use of these evolving reward techniques, we must answer the following fundamental question precisely:

*How fast can the reward change while still ensuring the convergence of an RL algorithm?*

This paper aims to establish a theoretical foundation for this setting by providing the first finite-time convergence analysis for an actor-critic algorithm with an evolving reward. We focus on a single-sample, single-timescale actor-critic algorithm with linear function approximation for the critic under Markovian sampling. This setting is particularly practical yet challenging, as the non-stationarity from the evolving reward affects both the policy gradient (actor) and the value estimation (critic), creating a complex feedback loop.

Specifically, we bound both the expected actor error and the expected critic error in terms of the number of iterations $T$ and the total change of the reward parameters. From this, we derive conditions on the evolving rate of reward parameters necessary to achieve asymptotic convergence and to maintain the $O(1/\sqrt{T})$ convergence rate as in the static-reward case. Moreover, it turns out that $O(1/\sqrt{T})$ convergence can be achieved if the reward parameter follows a gradient-based update with bounded gradient and the same timescale as the actor and the critic, providing theoretical guarantees to a wide range of practical techniques, including curiosity-driven reward shaping (Pathak et al., 2017), random network distillation methods (Burda et al., 2019), and soft actor-critic with automated entropy adjustment (Haarnoja et al., 2018b).

To handle the evolving reward, we exploit the Lipschitz continuity of the objective function and the optimal critic parameter with respect to the reward parameter, which relies on the Lipschitz continuity assumption for the reward itself. In addition, we introduce a novel analysis on the distribution mismatch caused by Markovian sampling, improving the convergence rate by a factor of $\log^2 T$ in the static-reward case.

In summary, our contributions include the following:

- **Important Problem Formulation:** We formalize the problem of Actor-Critic with Evolving Reward, where the reward parameter $\varphi_t$ (encompassing the true reward and regularization terms) can be updated by an arbitrary oracle at every time step.

- **Novel Non-Asymptotic Results:** Under standard assumptions (Linear function approximated critic, Lipschitz continuity of policy and reward, sufficient exploration), we derive the convergence rate of the single-sample single-timescale actor-critic algorithm with Markovian sampling, and show that it achieves a convergence rate of $O(1/\sqrt{T})$ to a neighborhood of a stationary point under mild condition on the evolving reward, validating a wide range of practical RL techniques.

- **Interesting Technical Tools:** We establish the necessary assumptions and key properties to analyze the effects of the evolving reward on standard actor-critic algorithms. We also provide a novel analysis of the distribution mismatch caused by Markovian sampling, which independently improves the convergence rate for the static-reward case.

## 2 RELATED WORK

Our work is developed upon the finite-time analysis of policy gradient methods and actor-critic methods.

**Finite-time analysis of Policy Gradient Methods.** The finite-time convergence guarantees of policy gradient methods have been studied by Agarwal et al. (2021); Mei et al. (2020); Xiao (2022), assuming access to exact gradient oracles. For the stochastic case where the algorithm can only access gradient estimators from sampled trajectories or transitions, convergence results in terms of sample complexity have been established by Liu et al. (2020); Ding et al. (2022; 2025); Fatkhullin et al. (2023); Mondal & Aggarwal (2024).

**Finite-time analysis of Actor-Critic Methods.** The finite-time analysis of actor-critic methods encompasses several variants of the algorithm, including the double-loop setting (Yang et al., 2019; Kumar et al., 2023; Xu et al., 2020a; Cayci et al., 2024; Gaur et al., 2024; Ganesh et al., 2025), the two-timescale setting (Wu et al., 2020; Xu et al., 2020b; Shen et al., 2023; Hong et al., 2023), and the single-timescale setting(Chen et al., 2021; Olshevsky & Gharesifard, 2023; Tian et al., 2023; Chen & Zhao, 2023; 2025). The analysis of single-timescale actor-critic methods is particularly relevant to our work. Both Chen et al. (2021) and Olshevsky & Gharesifard (2023) obtain an $O(1/\sqrt{T})$ convergence rate in discrete spaces, assuming i.i.d. sampling. Chen & Zhao (2023) and Tian et al. (2023) made efforts to tackle the more practical yet challenging Markovian sampling setting. However, the former considers the average-reward scenario instead of the commonly used discounted-reward scenario, while the latter employs Markovian samples for the critic and i.i.d. samples for the actor. Chen & Zhao (2025) ultimately resolves the problem of Markovian sampling, obtaining an $O(\log^2 T/\sqrt{T})$ convergence rate, and further extends the analysis to continuous action spaces. Additionally, Tian et al. (2023) incorporates an analysis of neural network approximated critics, while the other four assume a linear function approximated critic.

While we focus on analyzing the convergence of the single-timescale actor-critic algorithm under evolving rewards, we recognize two related research areas that focus on designing reinforcement learning (RL) algorithms for non-static Markov Decision Processes (MDPs), where both rewards and transitions can change:

**Adversarial RL and Non-stationary RL.** Tailored for online learning, adversarial RL (Even-Dar et al., 2009; Zimin & Neu, 2013; Jin et al., 2020) and non-stationary RL (Cheung et al., 2023; Feng et al., 2023; Mao et al., 2025) aim to minimize cumulative static or dynamic regret through strategic algorithm design in the presence of adversarial rewards and transitions. However, their formulation does not align with the practical scenarios we consider, where the RL policy is trained before executing, and the evolving reward techniques we previously mentioned focus on enhancing convergence during training and improving performance in execution, rather than adapting in an adversarial manner.

**Performative RL.** Performative RL (Mandal et al., 2023; Rank et al., 2024; Mandal & Radanovic, 2025) examines situations in which the underlying MDP changes in response to the deployed policy. This setting partially overlaps with ours, as performative rewards represent a specific case within our general evolving reward framework. Under certain mild assumptions regarding the performative rewards and transitions, these studies demonstrate the existence of a stable policy and provide finite-time convergence guarantees for their proposed algorithms, which involve iterating between deployment and retraining. However, their approach necessitates finding exact solutions for a saddle point at each retraining step, which is often infeasible in practice.

## 3 PRELIMINARIES

### 3.1 MARKOV DECISION PROCESS

We consider an infinite-horizon discounted Markov Decision Process (MDP), defined by the tuple $\mathcal{M} = (\mathcal{S}, \mathcal{A}, \mathcal{P}, r, \gamma)$. Here, $\mathcal{S}$ is the state space, $\mathcal{A}$ is the action space, $\mathcal{P} : \mathcal{S} \times \mathcal{A} \to \Delta(\mathcal{S})$ is the transition kernel, $r : \mathcal{S} \times \mathcal{A} \to \mathbb{R}$ is the reward function, and $\gamma \in (0, 1)$ is the discount factor.

A policy $\pi : \mathcal{S} \to \Delta(\mathcal{A})$ maps states to distributions over actions. Starting from an initial state $s_0$, at each time step $t = 0, 1, \cdots$, an action $a_t \sim \pi(\cdot|s_t)$ is sampled, yielding a reward $r_t = r(s_t, a_t)$ and transitioning to the next state $s_{t+1} \sim \mathcal{P}(\cdot|s_t, a_t)$. The standard value function $V^\pi(s)$ and action-value function $Q^\pi(s, a)$ represent the expected discounted cumulative reward starting from state $s$ (and action $a$), respectively, and are defined as

$$V^\pi(s) = \mathbb{E}\left[\sum_{t=0}^\infty \gamma^t r(s_t, a_t) \mid s_0 = s, a_t \sim \pi(\cdot|s_t), s_{t+1} \sim \mathcal{P}(\cdot|s_t, a_t)\right], \quad (1)$$

$$Q^\pi(s, a) = \mathbb{E}\left[\sum_{t=0}^\infty \gamma^t r(s_t, a_t) \mid s_0 = s, a_0 = a, s_{t+1} \sim \mathcal{P}(\cdot|s_t, a_t), a_{t+1} \sim \pi(\cdot|s_{t+1})\right]. \quad (2)$$

**Entropy Regularized MDPs.** Policy optimization often benefits from entropy regularization to encourage exploration (Haarnoja et al., 2017; 2018a;b). The entropy of a policy $\pi$ at a state $s$ is defined as $\mathcal{H}(\cdot|s) = -\int_{\mathcal{A}} \pi(a|s) \log \pi(a|s) \mathrm{d}a$. Let

$$H^\pi(s) = \mathbb{E}\left[\sum_{t=0}^\infty \gamma^t \mathcal{H}(\pi(\cdot|s_t)) \mid s_0 = s, a_t \sim \pi(\cdot|s_t), s_{t+1} \sim \mathcal{P}(\cdot|s_t, a_t)\right], \tag{3}$$

the regularized (soft) value function is defined as

$$\widetilde{V}^\pi(s) = V^\pi(s) + \alpha H^\pi(s),$$

where $\alpha \geq 0$ is a hyper-parameter, and $\alpha = 0$ recovers the standard, unregularized case. This formulation is equivalent to solving the original MDP with a regularized reward function:

$$\tilde{r}(s, a) = r(s, a) - \alpha \log \pi(a|s). \tag{4}$$

Consequently, $\widetilde{V}(s)$ and $\widetilde{Q}(s, a)$ can be interpreted as the value functions under this regularized reward $\tilde{r}$:

$$\widetilde{V}^\pi(s) = \mathbb{E}_{\pi, \mathcal{P}}\left[\sum_{t=0}^\infty \gamma^t \tilde{r}(s_t, a_t) \mid s_0 = s\right], \quad \widetilde{Q}^\pi(s, a) = \mathbb{E}_{\pi, \mathcal{P}}\left[\sum_{t=0}^\infty \gamma^t \tilde{r}(s_t, a_t) \mid s_0 = s, a_0 = a\right].$$

**Note on KL Regularization.** While we focus on entropy regularization for clarity, our analysis also applies to KL regularization against a fixed reference policy $\pi_{\text{ref}}$. The corresponding regularized reward becomes $\tilde{r}(s, a) = r(s, a) + \alpha \log \pi_{\text{ref}}(a|s) - \alpha \log \pi(a|s)$. Since the term $\alpha \log \pi_{\text{ref}}(a|s)$ can be absorbed into the base reward $r(s, a)$, we will consider only the entropy regularizer in the following analysis without loss of generality.

The goal of reinforcement learning is to find a parameterized policy $\pi_{\boldsymbol{\theta}}$ the maximizes the objective:

$$J(\boldsymbol{\theta}) = \int_{\mathcal{S}} \rho(s) \widetilde{V}^{\pi_{\boldsymbol{\theta}}}(s) \mathrm{d}s, \tag{5}$$

where $\boldsymbol{\theta}$ denotes the policy parameter and $\rho \in \Delta(\mathcal{S})$ is the initial distribution.

## 3.2 ACTOR-CRITIC METHOD

In order to optimize the policy parameter $\boldsymbol{\theta}$, a popular approach is to compute the gradient of the objective $J(\boldsymbol{\theta})$ and iteratively adjust $\boldsymbol{\theta}$ in the direction of $\nabla J(\boldsymbol{\theta})$. This approach, named the policy gradient method, is based on the policy gradient theorem (Sutton et al., 1999):

$$\nabla_{\boldsymbol{\theta}} J(\boldsymbol{\theta}) = \frac{1}{1-\gamma} \mathbb{E}_{s \sim \nu_\rho^{\pi_{\boldsymbol{\theta}}}(\cdot), a \sim \pi_{\boldsymbol{\theta}}(\cdot|s)} \left[\widetilde{Q}^{\pi_{\boldsymbol{\theta}}}(s, a) \nabla_{\boldsymbol{\theta}} \log \pi_{\boldsymbol{\theta}}(a|s)\right]$$

$$= \frac{1}{1-\gamma} \mathbb{E}_{s \sim \nu_\rho^{\pi_{\boldsymbol{\theta}}}(\cdot), a \sim \pi_{\boldsymbol{\theta}}(\cdot|s)} \left[\left(\widetilde{Q}^{\pi_{\boldsymbol{\theta}}}(s, a) - \widetilde{V}^{\pi_{\boldsymbol{\theta}}}(s)\right) \nabla_{\boldsymbol{\theta}} \log \pi_{\boldsymbol{\theta}}(a|s)\right], \tag{6}$$

where $\nu_\rho^{\pi_{\boldsymbol{\theta}}} \in \Delta(\mathcal{S})$ is the discounted visitation distribution defined as

$$\nu_\rho^{\pi_{\boldsymbol{\theta}}}(s) = (1-\gamma) \sum_{t=0}^\infty \gamma^t \Pr\left(s_t = s | s_0 \sim \rho(\cdot), a_t \sim \pi_{\boldsymbol{\theta}}(\cdot|s_t), s_{t+1} \sim \mathcal{P}(\cdot|s_t, a_t)\right).$$

Computing this gradient requires an estimator for the $\widetilde{V}$ function or $\widetilde{Q}$ function associated with the policy $\pi_{\boldsymbol{\theta}}$. A Monte-Carlo estimator (used by REINFORCE (Williams, 1992)) suffers from high variance, resulting in slow convergence. Hence, the actor-critic method (Konda & Tsitsiklis, 1999) introduces another trainable model to approximate the true value functions.

Following the setting of prior work (Chen et al., 2021; Olshevsky & Gharesifard, 2023; Chen & Zhao, 2023; 2025), we assume that the critic approximates the regularized value function linearly as

$$\widehat{V}_{\boldsymbol{\omega}}(s) = \phi(s)^\top \boldsymbol{\omega},$$

where $\phi : \mathcal{S} \to \mathbb{R}^d$ is a known feature mapping satisfying $\|\phi(s)\|_2 \le 1$ for any state $s$, and $\boldsymbol{\omega} \in \mathbb{R}^d$ is the trainable parameter. Note that $\widetilde{Q}^\pi(s,a) = \tilde{r}(s,a) + \gamma \mathbb{E}_{s'}[\widetilde{V}^\pi(s')]$, then with $s' \sim \mathcal{P}(\cdot|s,a)$, the temporal difference (TD) error

$$\hat{\delta}(s,a,s') = r(s,a) - \alpha \log \pi_{\boldsymbol{\theta}}(a|s) + (\gamma\phi(s') - \phi(s))^\top \boldsymbol{\omega} \tag{7}$$

serves as a biased but low-variance gradient estimator for (6), and the update rule for $\boldsymbol{\theta}$ will be

$$\boldsymbol{\theta}_{t+1} \leftarrow \boldsymbol{\theta}_t + \eta_t^\theta \hat{\delta}(s_t,a_t,s_t') \nabla_{\boldsymbol{\theta}} \log \pi_{\boldsymbol{\theta}}(a_t|s_t), \tag{8}$$

where $\eta_t^\theta$ denotes the step size for $\boldsymbol{\theta}$ at step $t$.

Also, we need to align $\widehat{V}_{\boldsymbol{\omega}}$ with the true value function $\widetilde{V}^{\pi_{\boldsymbol{\theta}}}$. As proven by Haarnoja et al. (2017), $\widetilde{V}^\pi$ is the unique solution to the soft Bellman equation $V = \mathcal{T}_\alpha^\pi V$ where the soft Bellman operator $\mathcal{T}_\alpha^\pi$ is defined as

$$\mathcal{T}_\alpha^\pi V(s) = \mathbb{E}_{a\sim\pi(\cdot|s), s'\sim\mathcal{P}(\cdot|a,s)}[r(s,a) - \alpha \log \pi(a|s) + \gamma V(s')].$$

Hence, a common practice is to adjust the value iteration update $V \leftarrow \mathcal{T}_\alpha^\pi V$ into a stochastic semi-gradient TD(0) update:

$$\boldsymbol{\omega}_{t+1} \leftarrow \boldsymbol{\omega}_t + \eta_t^\omega \hat{\delta}(s_t,a_t,s_t') \phi(s_t), \tag{9}$$

where $\eta_t^\omega$ denotes the step size for $\boldsymbol{\omega}$ at step $t$.

**Markovian Sampling.** In our single-sample, single-timescale setting, both actor and critic are updated using the same sample tuple $(s_t, a_t, s_t')$ at each step. Ideally, $s_t$ shall be sampled from the stationary distribution $\nu_\rho^{\pi_{\boldsymbol{\theta}}}$, but this is impractical. Instead, we adopt a Markovian sampling scheme (Chen & Zhao, 2025):

$$s_t \sim \widehat{\mathcal{P}}(\cdot|s_{t-1},a_{t-1}), a_t \sim \pi_{\boldsymbol{\theta}_t}(\cdot|s_t), s_t' \sim \mathcal{P}(\cdot|s_t,a_t),$$

where the sampling kernel $\widehat{\mathcal{P}}(\cdot|s,a) = \gamma\mathcal{P}(\cdot|s,a) + (1-\gamma)\rho(\cdot)$ ensures ergodicity. Let $\hat{\nu}_t(\cdot)$ denote the probability distribution of $s_t$ induced by this process. When the policy is fixed, $\hat{\nu}_t$ will converge to $\nu_\rho^{\pi_{\boldsymbol{\theta}}}$ geometrically. Under a slowly changing policy, the distribution mismatch $\|\hat{\nu}_t - \nu_\rho^{\pi_{\boldsymbol{\theta}_t}}\|_1$ can be controlled by the magnitude of the policy updates. Note that this constitutes an off-policy learning setting for the critic.

### 3.3 Actor-Critic with Evolving Reward

A central focus of this work is the setting where the regularized reward $\tilde{r}(s,a)$ evolves during training. Depending on the algorithmic design, this evolution can arise from modifications to the base reward $r(s,a)$, the regularization factor $\alpha$, or the policy $\pi_{\boldsymbol{\theta}}$ itself. To encompass these variables, we introduce a general reward parameter $\boldsymbol{\varphi}$ to include all factors that determine $\tilde{r}(s,a)$ along with $\boldsymbol{\theta}$, i.e.,

$$\tilde{r}_{\boldsymbol{\varphi},\boldsymbol{\theta}}(s,a) = r(s,a;\boldsymbol{\varphi}) - \alpha(\boldsymbol{\varphi}) \log \pi_{\boldsymbol{\theta}}(a|s).$$

We denote the corresponding soft value function as $\widetilde{V}_{\boldsymbol{\varphi}}^{\pi_{\boldsymbol{\theta}}}(s)$, and the policy objective as $J_{\boldsymbol{\varphi}}(\boldsymbol{\theta})$. The reward parameter $\boldsymbol{\varphi}$ is updated concurrently with $\boldsymbol{\theta}$ and $\boldsymbol{\omega}$ at each time step via an arbitrary update rule, which can either be a pre-defined schedule or a feedback-driven strategy. For the critic update, we also introduce a projection $\mathbf{Proj}_{C_{\boldsymbol{\omega}}}$ to keep the critic norm bounded by $C_{\boldsymbol{\omega}}$, which is widely adopted in the literature (Wu et al., 2020; Chen et al., 2021; Olshevsky & Gharesifard, 2023; Chen & Zhao, 2023; 2025). This framework, summarized in Algorithm 1, unifies a wide range of existing techniques, from automated reward shaping (Martin et al., 2017; Pathak et al., 2017; Burda et al., 2019) to adaptive entropy and KL regularization (Haarnoja et al., 2018b). We provide a further literature review of these eolving reward techniques in Appendix A.

## 4 Main Results

This section presents the finite-time convergence guarantees for the Actor-Critic with Evolving Reward algorithm (Algorithm 1). We begin by stating the standard assumptions required for our analysis, then present the main theorem and a key corollary. Finally, we provide an intuitive proof sketch to elucidate the key technical challenges and innovations.

---

**Algorithm 1** Actor Critic with Evolving Reward

---

Initialize: $\boldsymbol{\theta}_0, \boldsymbol{\omega}_0, \boldsymbol{\varphi}_0, \rho, \{\eta_t^\theta\}_{t \geq 0}, \{\eta_t^\omega\}_{t \geq 0}$
Sample $s_0 \sim \rho$
**for** $t = 0, 1, \cdots, T - 1$ **do**
    Sample $a_t \sim \pi_{\boldsymbol{\theta}_t}(\cdot|s_t), s_t' \sim \mathcal{P}(\cdot|s_t, a_t), s_{t+1} \sim \widehat{\mathcal{P}}(\cdot|s_t, a_t)$
    $\hat{\delta}_t \leftarrow \tilde{r}_{\boldsymbol{\varphi}_t, \boldsymbol{\theta}_t}(s_t, a_t) + (\gamma\phi(s_t') - \phi(s_t))^\top \boldsymbol{\omega}_t$
    $\boldsymbol{\theta}_{t+1} \leftarrow \boldsymbol{\theta}_t + \eta_t^\theta \hat{\delta}_t \nabla_{\boldsymbol{\theta}} \log \pi_{\boldsymbol{\theta}}(a_t|s_t)$
    $\boldsymbol{\omega}_{t+1} \leftarrow \mathbf{Proj}_{C_{\boldsymbol{\omega}}} \left( \boldsymbol{\omega}_t + \eta_t^\omega \hat{\delta}_t \phi(s_t) \right)$
    $\boldsymbol{\varphi}_{t+1} \leftarrow \mathbf{UpdateReward}(\boldsymbol{\varphi_t})$
**end for**

---

### 4.1 ASSUMPTIONS

Our analysis relies on several standard assumptions in the literature, which we adapt to accommodate the evolving reward setting.

By taking the expectation of $\boldsymbol{\omega}_{t+1}$ conditioning on $\boldsymbol{\omega}_t$ in (9) with respect to the discounted visitation distribution, we have

$$\mathbb{E}[\boldsymbol{\omega}_{t+1}|\boldsymbol{\omega}_t] = \boldsymbol{\omega}_t + \eta_t^\omega(\boldsymbol{b}_{\boldsymbol{\varphi},\boldsymbol{\theta}} - \boldsymbol{A}_{\boldsymbol{\theta}}\boldsymbol{\omega}_t),$$

where

$$\boldsymbol{A}_{\boldsymbol{\theta}} = \mathbb{E}_{s \sim \nu_\rho^{\pi_{\boldsymbol{\theta}}}(\cdot), a \sim \pi_{\boldsymbol{\theta}}(\cdot|s), s' \sim \mathcal{P}(\cdot|s,a)} \left[ \phi(s)(\phi(s) - \gamma\phi(s'))^\top \right], \tag{10}$$

$$\boldsymbol{b}_{\boldsymbol{\varphi},\boldsymbol{\theta}} = \mathbb{E}_{s \sim \nu_\rho^{\pi_{\boldsymbol{\theta}}}(\cdot), a \sim \pi_{\boldsymbol{\theta}}(\cdot|s)} \left[ \tilde{r}_{\boldsymbol{\varphi},\boldsymbol{\theta}}(s, a)\phi(s) \right]. \tag{11}$$

It has been shown by Sutton et al. (1998) that the TD limiting point $\boldsymbol{\omega}^*(\boldsymbol{\varphi}, \boldsymbol{\theta})$ satisfies

$$\boldsymbol{A}_{\boldsymbol{\theta}}\boldsymbol{\omega}^*(\boldsymbol{\varphi}, \boldsymbol{\theta}) = \boldsymbol{b}_{\boldsymbol{\varphi},\boldsymbol{\theta}}. \tag{12}$$

To ensure the existence of $\boldsymbol{\omega}^*$, we need $A_{\boldsymbol{\theta}}$ to be non-singular. Fortunately, we can show that $A_{\boldsymbol{\theta}}$ is positive definite given sufficient exploration over the state space.

**Assumption 4.1 (Sufficient Exploration)** *Let* $\Sigma_{\boldsymbol{\theta}} = \mathbb{E}_{\nu_\rho^{\pi_{\boldsymbol{\theta}}}} \left[ \phi(s)\phi(s)^\top \right]$, *then for any* $\boldsymbol{\theta} \in \Omega(\boldsymbol{\theta})$, $\Sigma_{\boldsymbol{\theta}}$ *is positive definite with singular values lower-bounded by* $\lambda_\Sigma > 0$.

**Proposition 4.2** *For any* $\boldsymbol{\theta} \in \Omega(\boldsymbol{\theta})$, $A_{\boldsymbol{\theta}}$ *is positive definite with singular values lower-bounded by* $\lambda = (1 - \sqrt{\gamma})\lambda_\Sigma$.

Proposition 4.2 is widely adopted as an assumption in analyzing TD-learning and actor-critic with linear function approximation (Wu et al., 2020; Chen et al., 2021; Olshevsky & Gharesifard, 2023; Chen & Zhao, 2023; 2025). Although the definition of $A_{\boldsymbol{\theta}}$ here differs from previous literature where the expectation is taken over the stationary distribution instead of the discounted visitation distribution as in our definition, this nice property can still be induced from the fundamental Assumption 4.1 (Bhandari et al., 2018). Equipped with Proposition 4.2, we can now solve from (12) that $\boldsymbol{\omega}^*(\boldsymbol{\varphi}, \boldsymbol{\theta}) = \boldsymbol{A}_{\boldsymbol{\theta}}^{-1} \boldsymbol{b}_{\boldsymbol{\varphi},\boldsymbol{\theta}}$, and further imply that $\boldsymbol{\omega}^*(\boldsymbol{\varphi}, \boldsymbol{\theta})$ is bounded by some constant $C_{\boldsymbol{\omega}}$ since both $\boldsymbol{A}_{\boldsymbol{\theta}}^{-1}$ and $\boldsymbol{b}_{\boldsymbol{\varphi},\boldsymbol{\theta}}$ can be shown bounded, which justifies the projection operator introduced in Algorithm 1.

As $\boldsymbol{\omega}^*$ is bounded, $\widehat{V}_{\boldsymbol{\omega}^*}$ is bounded, the linear function approximation error has a uniform upper bound, denoted as $\epsilon$. Formally,

$$\epsilon := \sup_{\boldsymbol{\theta},\boldsymbol{\varphi}} \sqrt{\mathbb{E}_{s \sim \nu_\rho^{\pi_{\boldsymbol{\theta}}}(\cdot)} \left[ \left( \phi(s)^\top \boldsymbol{\omega}^*(\boldsymbol{\varphi}, \boldsymbol{\theta}) - \widetilde{V}^{\pi_{\boldsymbol{\theta}}}(s) \right)^2 \right]}. \tag{13}$$

The error $\epsilon$ is zero if $\widetilde{V}^{\pi_{\boldsymbol{\theta}}}(\cdot)$ is indeed a linear function for any $\boldsymbol{\varphi}$ and $\boldsymbol{\theta}$ given the feature mapping $\phi(\cdot)$. To capture the bias of the TD-gradient estimator for the actor, we need the following bound that controls the error of TD-errors (refer to Appendix D for a detailed proof):

**Proposition 4.3** *For any $\boldsymbol{\theta} \in \Omega(\boldsymbol{\theta})$, $\boldsymbol{\varphi} \in \Omega(\boldsymbol{\varphi})$,*

$$\sqrt{\mathbb{E}_{\nu_\rho^{\pi_{\boldsymbol{\theta}}}, \pi_{\boldsymbol{\theta}}, \mathcal{P}} \left[ \left( \left( \gamma \widehat{V}_{\boldsymbol{\omega}^*}(s') - \widehat{V}_{\boldsymbol{\omega}^*}(s) \right) - \left( \gamma \widetilde{V}^{\pi_{\boldsymbol{\theta}}}(s') - \widetilde{V}^{\pi_{\boldsymbol{\theta}}}(s) \right) \right)^2 \right]} \leq 2\sqrt{2}\epsilon.$$

**Assumption 4.4 (Lipschitz Continuity of Policy)** *There exist constants $L$ and $S$ such that for any $\boldsymbol{\theta} \in \Omega(\boldsymbol{\theta}), s \in \mathcal{S}, a \in \mathcal{A}$,*

$$\|\nabla_{\boldsymbol{\theta}} \log \pi_{\boldsymbol{\theta}}(a|s)\|_2 \leq L, \quad \|\nabla_{\boldsymbol{\theta}}^2 \log \pi_{\boldsymbol{\theta}}(a|s)\|_2 \leq S.$$

Assumption 4.4 is standard in the literature of policy gradient and actor-critic (Wu et al., 2020; Chen et al., 2021; Olshevsky & Gharesifard, 2023; Chen & Zhao, 2023; Tian et al., 2023), which further implies the following proposition that the policy $\pi_{\boldsymbol{\theta}}$ is Lipschitz continuous with respect to $\boldsymbol{\theta}$ (refer to Appendix D for a detailed proof):

**Proposition 4.5** *For any $\boldsymbol{\theta}_1, \boldsymbol{\theta}_2 \in \Omega(\boldsymbol{\theta}), s \in \mathcal{S}$, $\|\pi_{\boldsymbol{\theta}_1}(\cdot|s) - \pi_{\boldsymbol{\theta}_2}(\cdot|s)\|_1 \leq L\|\boldsymbol{\theta}_1 - \boldsymbol{\theta}_2\|_2$.*

Apart from the policy, we also need the Lipschitz continuity for the regularized reward to control the bias caused by the evolving reward.

**Assumption 4.6 (Lipschitz Continuity of Regularized Reward)** *There exist constants $C, D > 0$ such that for any $\boldsymbol{\theta} \in \Omega(\boldsymbol{\theta})$, $\boldsymbol{\varphi} \in \Omega(\boldsymbol{\varphi})$, and $s \in \mathcal{S}$, the expected regularized reward satisfies:*

1. ***Boundedness:*** $\left| \mathbb{E}_{a \sim \pi_{\boldsymbol{\theta}}(\cdot|s)}[\tilde{r}_{\boldsymbol{\varphi}, \boldsymbol{\theta}}(s, a)] \right| \leq C$

2. ***Bounded Variance:*** $\mathbb{E}_{a \sim \pi_{\boldsymbol{\theta}}(\cdot|s)}[\tilde{r}_{\boldsymbol{\varphi}, \boldsymbol{\theta}}(s, a)^2] \leq C^2$

3. ***Lipschitz in $\boldsymbol{\theta}$:*** $\|\nabla_{\boldsymbol{\theta}} \mathbb{E}_{a \sim \pi_{\boldsymbol{\theta}}(\cdot|s)}[\tilde{r}_{\boldsymbol{\varphi}, \boldsymbol{\theta}}(s, a)]\|_2 \leq CL$

4. ***Smoothness in $\boldsymbol{\theta}$:*** $\|\nabla_{\boldsymbol{\theta}}^2 \mathbb{E}_{a \sim \pi_{\boldsymbol{\theta}}(\cdot|s)}[\tilde{r}_{\boldsymbol{\varphi}, \boldsymbol{\theta}}(s, a)]\|_2 \leq C(L^2 + S)$

5. ***Lipschitz in $\boldsymbol{\varphi}$:*** $\|\nabla_{\boldsymbol{\varphi}} \mathbb{E}_{a \sim \pi_{\boldsymbol{\theta}}(\cdot|s)}[\tilde{r}_{\boldsymbol{\varphi}, \boldsymbol{\theta}}(s, a)]\|_2 \leq D$

**Parts 1 and 2** of Assumption 4.6 are natural extensions of the standard bounded-reward assumption to the expected regularized reward. The expectation over actions is necessary because the entropy regularization term $-\alpha \log \pi_{\boldsymbol{\theta}}(a|s)$ can be unbounded for individual actions, but its expectation $\alpha \mathcal{H}(\pi_{\boldsymbol{\theta}}(\cdot|s))$ is bounded for finite action spaces. For continuous action spaces, entropy regularization implicitly constrains the policy to have bounded entropy, as unbounded entropy would lead to infinite negative rewards, which is practically avoided.

**Parts 3 and 4** of Assumption 4.6 naturally follow from Part 1 and Assumption 4.4. Since

$$\nabla_{\boldsymbol{\theta}} \mathbb{E}_{a \sim \pi_{\boldsymbol{\theta}}(\cdot|s)}[\tilde{r}_{\boldsymbol{\varphi}, \boldsymbol{\theta}}(s, a)] = \mathbb{E}_{a \sim \pi_{\boldsymbol{\theta}}(\cdot|s)}[\tilde{r}_{\boldsymbol{\varphi}, \boldsymbol{\theta}}(s, a) \nabla_{\boldsymbol{\theta}} \log \pi_{\boldsymbol{\theta}}(a|s)],$$

$$\nabla_{\boldsymbol{\theta}}^2 \mathbb{E}_{a \sim \pi_{\boldsymbol{\theta}}(\cdot|s)}[\tilde{r}_{\boldsymbol{\varphi}, \boldsymbol{\theta}}(s, a)] = \mathbb{E}_{a \sim \pi_{\boldsymbol{\theta}}(\cdot|s)}[(\tilde{r}_{\boldsymbol{\varphi}, \boldsymbol{\theta}}(s, a) - \alpha) \nabla_{\boldsymbol{\theta}} \log \pi_{\boldsymbol{\theta}}(a|s) \nabla_{\boldsymbol{\theta}} \log \pi_{\boldsymbol{\theta}}(a|s)^\top]$$
$$+ \mathbb{E}_{a \sim \pi_{\boldsymbol{\theta}}(\cdot|s)}[\tilde{r}_{\boldsymbol{\varphi}, \boldsymbol{\theta}}(s, a) \nabla_{\boldsymbol{\theta}}^2 \log \pi_{\boldsymbol{\theta}}(a|s)],$$

the Lipschitz continuity and smoothness w.r.t. $\boldsymbol{\theta}$ can be derived from the boundedeness of $\mathbb{E}_{a \sim \pi_{\boldsymbol{\theta}}(\cdot|s)}[\tilde{r}_{\boldsymbol{\varphi}, \boldsymbol{\theta}}(s, a)]$, $\nabla_{\boldsymbol{\theta}} \log \pi_{\boldsymbol{\theta}}(a|s)$, $\nabla_{\boldsymbol{\theta}}^2 \log \pi_{\boldsymbol{\theta}}(a|s)$ and $\alpha$.

**Part 5** of Assumption 4.6 is the most critical for handling evolving rewards. It guarantees that small changes in the reward parameters $\boldsymbol{\varphi}$ (e.g., from reward shaping or entropy adjustment) lead to proportionally small changes in the expected reward. This allows the algorithm to track the evolving learning objective rather than being destabilized by it.

In essence, Assumption 4.6 ensures that the regularized reward function changes in a controlled and predictable manner as the policy and reward parameters evolve. This is crucial for analyzing non-stationary learning dynamics.

## 4.2 MAIN RESULTS

With the assumptions above, we are ready to present our finite-time analysis of Algorithm 1. We measure the performance of Algorithm 1 using the following time-averaged errors over the second half of the $T$ iterations:

- **Actor Error:** $G_T = \frac{1}{T/2} \sum_{t=T/2}^{T-1} \mathbb{E} \left\| \nabla_{\boldsymbol{\theta}} J_{\boldsymbol{\varphi}_t}(\boldsymbol{\theta}_t) \right\|_2^2$

- **Critic Error:** $W_T = \frac{1}{T/2} \sum_{t=T/2}^{T-1} \mathbb{E} \|\boldsymbol{\omega}_t - \boldsymbol{\omega}_t^*\|_2^2$, where $\boldsymbol{\omega}_t^* = \boldsymbol{\omega}^*(\boldsymbol{\varphi}_t, \boldsymbol{\theta}_t)$

- **Reward Variation:** $F_T = \frac{1}{T/2} \sum_{t=T/2}^{T-1} \mathbb{E} \left\| \boldsymbol{\varphi}_{t+1} - \boldsymbol{\varphi}_t \right\|_2^2$

**Theorem 4.7** *Consider Algorithm 1 with $\eta_t^\theta = \frac{c_\theta}{\sqrt{t}}$ and $\eta_t^\omega = \frac{c_\omega}{\sqrt{t}}$, where the ratio $\frac{c_\theta}{c_\omega}$ is chosen to be sufficiently small such that $\frac{c_\theta}{c_\omega} \leq \frac{\lambda}{LS_\omega} \wedge \frac{1}{16LL_\omega}$. Under Assumption 4.1, 4.4 and 4.6, the following bounds hold:*

$$G_T = O\left(\frac{1}{\sqrt{T}}\right) + O\left(F_T \sqrt{T}\right) + O\left(\sqrt{\frac{F_T}{T}}\right) + O(\epsilon)$$

$$W_T = O\left(\frac{1}{\sqrt{T}}\right) + O\left(F_T \sqrt{T}\right) + O\left(\sqrt{\frac{F_T}{T}}\right) + O(\epsilon)$$

**Interpretation of Theorem 4.7:**

- **Static-Reward Case** ($F_T \equiv 0$): The algorithm achieves the canonical $O(1/\sqrt{T})$ convergence rate for both actor and critic, matching the best-known rate for single-timescale actor-critic methods under i.i.d. sampling (Chen et al., 2021; Olshevsky & Gharesifard, 2023). Our analysis, by carefully handling the Markovian sampling, improves upon previous works by eliminating a $\log^2 T$ factor compared to Tian et al. (2023); Chen & Zhao (2023; 2025).

- **Evolving-Reward Case** ($F_T > 0$): The convergence rate depends critically on the total variation of the reward parameters, $F_T$. For the errors to converge to zero asymptotically, we require $F_T = o\left(1/\sqrt{T}\right)$. To preserve the $O(1/\sqrt{T})$ rate, we need the stronger condition $F_T = O\left(1/T\right)$. This means the reward function must change slowly enough for the actor-critic algorithm to track it effectively.

The following corollary shows that a common class of reward update rules satisfies this stringent condition.

**Corollary 4.8** *If the reward parameter adopts a gradient-based update rule, i.e.*

$$\boldsymbol{\varphi}_{t+1} \leftarrow \boldsymbol{\varphi}_t + \eta_t^\varphi h_\varphi(t),$$

*then given $\mathbb{E}\|h_\varphi(t)\|_2^2 \leq C_{\boldsymbol{\varphi}}^2$ and $\eta_t^\varphi = \frac{c_{\boldsymbol{\varphi}}}{\sqrt{t}}$ where $C_{\boldsymbol{\varphi}}$ and $c_{\boldsymbol{\varphi}}$ are constants, we have $F_K = O\left(\frac{1}{T}\right)$, and hence*

$$G_T = O\left(\frac{1}{\sqrt{T}}\right) + O(\epsilon), \quad W_T = O\left(\frac{1}{\sqrt{T}}\right) + O(\epsilon).$$

The proof of Corollary 4.8 can be found in Appendix D. Here, the step size for updating the reward parameter is of the same order as the actor's and the critic's, and hence the requirements can be achieved by applying gradient clipping, a technique that is very common in practice. Therefore, Corollary 4.8 provides a solid theoretical foundation for a wide range of empirical practice of RL.

### 4.3 PROOF SKETCH OF THE MAIN THEOREM

The proof of Theorem 4.7 proceeds in three interconnected steps, which we outline below. A rigorous proof is provided in Appendix C. The key innovations lie in (1) rigorously analyzing the impact of evolving rewards by establishing Lipschitz continuity properties of policy objective $J_{\boldsymbol{\varphi}}(\boldsymbol{\theta})$ and optimal critic parameter $\boldsymbol{\omega}^*(\boldsymbol{\varphi}, \boldsymbol{\theta})$ w.r.t $\varphi$, and (2) providing a novel analysis on the distribution mismatch induced by Markovian sampling through deriving the following key proposition (refer to Appendix D for a detailed proof):

**Proposition 4.9** *Following the Markovian sampling strategy described in Algorithm 1, we have*

$$\mathbb{E}\|\hat{\nu}_t - \nu_\rho^{\pi_{\boldsymbol{\theta}_t}}\|_1 \leq LC_\delta L_\nu \sum_{k=0}^{t-1} \gamma^{t-1-k}\eta_k^\theta + \gamma^t\|\rho - \nu_\rho^{\pi_{\boldsymbol{\theta}_0}}\|_1$$

*for any $t \geq 0$, where $C_\delta$ and $L_\nu$ are constants (refer to Appendix B for a formal definition).*

This analysis does not rely on the mixing time of the ergodic Markov chain. Instead, it directly utilizes the contraction properties of the induced operator acting on state distributions, which is a stronger property than the ergodicity, hence resulting in a tighter bound on the distribution mismatch.

**Step 1: Bounding the Actor Error.** The primary challenge introduced by an evolving reward is that the policy optimization objective $J_{\boldsymbol{\varphi}_t}(\boldsymbol{\theta}_t)$ changes at every time step $t$. To address this, we first show that $J_{\boldsymbol{\varphi}}(\boldsymbol{\theta})$ is $D_J$-Lipschitz with respect to the reward parameter $\boldsymbol{\varphi}$ (Lemma B.1), which allows us to bound the change in the objective function by the change in $\boldsymbol{\varphi}$:

$$\mathbb{E}[J_{\boldsymbol{\varphi}_{t+1}}(\boldsymbol{\theta}_{t+1})] - J_{\boldsymbol{\varphi}_t}(\boldsymbol{\theta}_t) \geq -D_J\mathbb{E}\|\boldsymbol{\varphi}_{t+1} - \boldsymbol{\varphi}_t\|_2 + \mathbb{E}[J_{\boldsymbol{\varphi}_t}(\boldsymbol{\theta}_{t+1})] - J_{\boldsymbol{\varphi}_t}(\boldsymbol{\theta}_t)$$

We then analyze the improvement in the objective for a fixed reward parameter. A Taylor expansion of $J_{\boldsymbol{\varphi}_t}(\boldsymbol{\theta})$ around $\boldsymbol{\theta}_t$ yields a bound on the squared policy gradient norm, $\|\nabla_{\boldsymbol{\theta}} J_{\boldsymbol{\varphi}_t}(\boldsymbol{\theta}_t)\|_2^2$. This bound involves several error terms:

- $I_1$(**Approximation Error**): This term arises from the bias introduced by linear function approximation and is bounded by $O(\epsilon)$.
- $I_2$(**Critic Error**): This term captures the error from using an estimated critic $\boldsymbol{\omega}_t$ instead of the optimal critic $\boldsymbol{\omega}_t^*$ and is bounded by $O(\|\boldsymbol{\omega}_t - \boldsymbol{\omega}_t^*\|_2)$.
- $I_3$(**Markovian Noise**): This term quantifies the error due to sampling states from the Markovian distribution $\hat{\nu}_t$ rather than the true stationary distribution $\nu_\rho^{\pi_{\boldsymbol{\theta}_t}}$. Proposition 4.9 provides a tighter bound on this distribution mismatch, which is crucial for improving the overall convergence rate.

After summing over iterations and applying a telescoping series, we obtain the following inequality for the actor error (Theorem C.1):

$$(1-\gamma)G_T \leq 2L\sqrt{G_T W_T} + O\left(\sqrt{\frac{F_T}{T}}\right) + O\left(\frac{1}{\sqrt{T}}\right) + O(\epsilon) \tag{14}$$

**Step 2: Bounding the Critic Error.** The critic update must track a moving target: the optimal parameter $\boldsymbol{\omega}_t^* = \boldsymbol{\omega}^*(\boldsymbol{\varphi}_t, \boldsymbol{\theta}_t)$ changes with both the policy parameter $\boldsymbol{\theta}_t$ and the reward parameter $\boldsymbol{\varphi}_t$. We analyze the evolution of the critic error $\|\boldsymbol{\omega}_{t+1} - \boldsymbol{\omega}_{t+1}^*\|_2^2$. A central decomposition yields:

$$\mathbb{E}\|\boldsymbol{\omega}_{t+1} - \boldsymbol{\omega}_{t+1}^*\|_2^2 \leq \|\boldsymbol{\omega}_t - \boldsymbol{\omega}_t^*\|_2^2 + 2\mathbb{E}\left\|\hat{\delta}(s_t, a_t, s_t')\phi(s_t)\right\|_2^2 + 2\mathbb{E}\|\boldsymbol{\omega}_t^* - \boldsymbol{\omega}_{t+1}^*\|_2^2$$

$$+ 2\eta_t^\omega\mathbb{E}\left\langle\boldsymbol{\omega}_t - \boldsymbol{\omega}_t^*, \hat{\delta}(s_t, a_t, s_t')\phi(s_t)\right\rangle + 2\mathbb{E}\left\langle\boldsymbol{\omega}_t - \boldsymbol{\omega}_t^*, \boldsymbol{\omega}_t^* - \boldsymbol{\omega}_{t+1}^*\right\rangle,$$

where $\mathbb{E}\left\langle\boldsymbol{\omega}_t - \boldsymbol{\omega}_t^*, \hat{\delta}(s_t, a_t, s_t')\phi(s_t)\right\rangle$ is further decomposed into three components:

- $J_1$: This term is zero by the definition of $\boldsymbol{\omega}_t^*$.
- $J_2$(**Contraction**): This term provides a negative contribution $-\lambda\|\boldsymbol{\omega}_t - \boldsymbol{\omega}_t^*\|_2^2$, ensuring the critic error contracts towards zero.
- $J_3$(**Markovian Noise**): Similar to $I_3$ in the actor analysis, this term is bounded using Proposition 4.9.

The critical difference from the static-reward case is the presence of terms involving $\boldsymbol{\omega}_t^* - \boldsymbol{\omega}_{t+1}^*$. We bound these by establishing the Lipschitz continuity of $\boldsymbol{\omega}^*$ with respect to both $\boldsymbol{\theta}$ and $\boldsymbol{\varphi}$ (Lemma B.5). This introduces terms proportional to $\mathbb{E}\|\boldsymbol{\varphi}_{t+1} - \boldsymbol{\varphi}_t\|_2^2$ and $\mathbb{E}\|\boldsymbol{\varphi}_{t+1} - \boldsymbol{\varphi}_t\|_2$ into the bound. After summation, we derive the following inequality for the critic error (Theorem C.2):

$$\frac{1}{1-\gamma}W_T \leq 2L_\omega\frac{c_{\boldsymbol{\theta}}}{c_{\boldsymbol{\omega}}}\sqrt{G_T W_T} + O\left(F_T\sqrt{T}\right) + O\left(\sqrt{\frac{F_T}{T}}\right) + O\left(\frac{1}{\sqrt{T}}\right) + O(\epsilon) \tag{15}$$

**Step 3: Solving the System of Inequalities**  Steps 1 and 2 result in a system of two inequalities (14 and 15) that couple the actor error $G_T$ and the critic error $W_T$. To solve this system, we use the algebraic inequality

$$2\sqrt{G_T W_T} \leq \frac{1-\gamma}{2L} G_T + \frac{2L}{1-\gamma} W_T.$$

By substituting this into the inequalities and choosing the step-size ratio $\frac{c_\theta}{c_\omega}$ to be sufficiently small, we can decouple the two errors and obtain a bound of $O(1/\sqrt{T}) + O(F_T\sqrt{T}) + O(\sqrt{F_T/T}) + O(\epsilon)$ for both $G_T$ and $W_T$, thus completing the proof.

## 5 CONCLUSION

In this work, we have undertaken a systematic theoretical investigation of actor-critic methods in the presence of evolving rewards—a setting that mirrors the reality of many practical RL algorithms but has been largely overlooked by theoretical analyses. We formulated the problem, established necessary assumptions, and provided the first finite-time convergence guarantees for a single-timescale actor-critic algorithm under Markovian sampling.

Our analysis demonstrates that the single-timescale actor-critic algorithm is remarkably robust to reward non-stationarity. The canonical $O(1/\sqrt{T})$ convergence rate can be maintained for both the actor and critic, provided the reward parameters evolve at a controlled pace. A key corollary confirms that gradient-based reward updates—a common pattern in algorithms that learn intrinsic rewards or adapt regularization strengths—satisfy this condition, thereby providing a solid theoretical foundation for their empirical success. Furthermore, our novel technique for bounding distribution mismatch under Markovian sampling yields a tighter analysis, improving upon prior rates by a factor of $\log^2 T$ even when the reward is static.

This work opens several avenues for future research. Extending the analysis to nonlinear function approximation, particularly with neural networks, is a critical next step. Furthermore, exploring the implications of our theoretical findings for the design of more effective and provably stable reward-shaping algorithms presents an exciting direction for both theoretical and applied work. Finally, this analysis lays a foundational stone for a deeper theoretical understanding of reinforcement learning with dynamic objectives due to evolving reward, shifting initial distribution or transition probabilities.

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

## A  LITERATURE REVIEW OF EVOLVING REWARD TECHNIQUES

The idea of modifying rewards to improve learning is long-standing. Potential-based reward shaping, introduced and developed by Ng et al. (1999); Wiewiora (2003); Asmuth et al. (2008); Devlin & Kudenko (2012), defines the shaped reward as $\gamma\Phi(s') - \Phi(s)$ where $\Phi(\cdot)$ is a potential function to guarantee policy invariance. More recent work, however, modifies the reward to balance the exploration and exploitation behavior of the policy. This sorts of work include randomly perturbing the reward function (Mahankali et al., 2024), various design of explicit exploration bonus like count-based methods (Martin et al., 2017; Machado et al., 2020; Lobel et al., 2023), curiosity-driven methods (Pathak et al., 2017; 2019; Ramesh et al., 2022; Sun et al., 2022) and random network distillation (Burda et al., 2019; Yang et al., 2024), as well as fully self-supervised intrinsic rewards (Zheng et al., 2018; Stadie et al., 2020; Memarian et al., 2021; Mguni et al., 2023; Ma et al., 2024) or incorporation of prior knowledge (Trott et al., 2019; Hu et al., 2020; Gupta et al., 2023) that enhance the performance of the resulting policy in terms of the original reward.

Another series of work that result in evolving rewards is the entropy or KL regularization. Entropy regularization (Haarnoja et al., 2018a;b; Ahmed et al., 2019) is commonly used technique to encourage exploration and avoid near-deterministic suboptimal policy. KL regularization is common in fine-tuning RL policies, especially in training Large Language Models (Ouyang et al., 2022; Shao

et al., 2024) and Diffusion Models (Fan et al., 2023). These methods result in evolving reward because the regularization term $-\alpha\mathcal{H}(\pi(\cdot|s))$ (or $-\alpha d_{\text{KL}}(\pi(\cdot|s)||\pi_{\text{ref}}(\cdot|s))$) is equivalent to a penalty term $-\alpha\log\pi(a|s)$ (or $-\alpha\log\frac{\pi(a|s)}{\pi_{\text{ref}}(a|s)}$) added to the reward function $r(s,a)$. Hence, the reward function will change because of the under-training policy $\pi(\cdot|s)$, the adaptive regularization factor $\alpha$, or the change of the reference policy $\pi_{\text{ref}}$.

Besides, curriculum learning (Narvekar et al., 2020) is also closely related, as it inherently involve a sequence of evolving learning objectives (and thus rewards).

## B  PRELIMINARY LEMMAS

**Lemma B.1** *There exist constants $C_J$, $L_J$, $S_J$ and $D_J$ such that for any $\boldsymbol{\theta}\in\Omega(\boldsymbol{\theta}), s\in\mathcal{S}$, $\widetilde{V}_{\boldsymbol{\varphi}}^{\pi_{\boldsymbol{\theta}}}(s)$ is $C_J$-bounded, $L_J$-Lipschitz and $S_J$-smooth w.r.t. $\boldsymbol{\theta}$, and $D_J$-Lipschitz w.r.t. $\varphi$, where $C_J = O((1-\gamma)^{-1})$, $L_J = O((1-\gamma)^{-2})$, $S_J = O((1-\gamma)^{-3})$, $D_J = O((1-\gamma)^{-1})$.*

**Corollary B.2** *There exist constants $L_\nu$ and $S_\nu$ such that for any $\boldsymbol{\theta}\in\Omega(\boldsymbol{\theta})$, $\nu_\rho^{\pi_{\boldsymbol{\theta}}}(\cdot)$ is $L_\nu$-Lipschitz and $S_\nu$-smooth w.r.t. $\boldsymbol{\theta}$ in terms of $\|\cdot\|_1$, where $L_\nu = O((1-\gamma)^{-1})$, $S_\nu = O((1-\gamma)^{-2})$.*

**Lemma B.3** *There exist constants $L_A$ and $S_A$ such that $A_{\boldsymbol{\theta}}$ is $L_A$-Lipschitz and $S_A$-smooth w.r.t $\boldsymbol{\theta}$, where $L_A = O((1-\gamma)^{-1})$, $S_A = O((1-\gamma)^{-2})$.*

**Lemma B.4** *There exist constants $C_b$, $L_b$, $S_b$ and $D_b$ such that $b_{\boldsymbol{\theta},\boldsymbol{\varphi}}$ is $C_b$-bounded, $L_b$-Lipschitz and $S_b$-smooth w.r.t $\boldsymbol{\theta}$ and $D_b$-Lipschitz w.r.t $\boldsymbol{\varphi}$, where $C_b = O(1)$, $L_b = O((1-\gamma)^{-1})$, $S_b = O((1-\gamma)^{-2})$, $D_b = O(1)$.*

**Lemma B.5** *There exist constants $C_{\boldsymbol{\omega}}$, $L_{\boldsymbol{\omega}}$ and $S_{\boldsymbol{\omega}}$ and $D_{\boldsymbol{\omega}}$ such that $\boldsymbol{\omega}^*(\boldsymbol{\varphi},\boldsymbol{\theta})$ is $C_{\boldsymbol{\omega}}$-bounded, $L_{\boldsymbol{\omega}}$-Lipschitz and $S_{\boldsymbol{\omega}}$-smooth w.r.t $\boldsymbol{\theta}$ and $D_{\boldsymbol{\omega}}$-Lipschitz w.r.t $\boldsymbol{\varphi}$, where $C_{\boldsymbol{\omega}} = O(\lambda^{-1})$, $L_{\boldsymbol{\omega}} = O((1-\gamma)^{-1}\lambda^{-2})$, $S_{\boldsymbol{\omega}} = O((1-\gamma)^{-2}\lambda^{-3})$, $D_{\boldsymbol{\omega}} = O(\lambda^{-1})$.*

**Lemma B.6** *There exists a constant $C_\delta$ such that for any $\boldsymbol{\theta}\in\Omega(\boldsymbol{\theta})$, $\boldsymbol{\varphi}\in\Omega(\boldsymbol{\varphi})$ and $\|\boldsymbol{\omega}\|_2 \le C_{\boldsymbol{\omega}}$,*

$$\mathbb{E}_{\nu,\pi_{\boldsymbol{\theta}},\mathcal{P}}[\hat{\delta}(s,a,s')^2] \le C_\delta^2,$$

*where $C_\delta = O(\lambda^{-1})$.*

## C  PROOF OF MAIN THEOREM

### C.1  STEP 1: BOUNDING THE ACTOR ERROR

**Theorem C.1 (Actor Update)** *Tate $\eta_t^\theta = \frac{c_{\boldsymbol{\theta}}}{\sqrt{t}}$ where $c_{\boldsymbol{\theta}}$ is a constant, then*

$$G_T \le \frac{2L}{1-\gamma}\sqrt{G_T W_T} + O\left(\sqrt{\frac{F_T}{T}}\right) + O\left(\frac{1}{\sqrt{T}}\right) + O(\epsilon).$$

*Proof*  We first bound the change of the objective function by the change of the reward parameter:

$$\mathbb{E}[J_{\boldsymbol{\varphi}_{t+1}}(\boldsymbol{\theta}_{t+1})] - J_{\boldsymbol{\varphi}_t}(\boldsymbol{\theta}_t) = \mathbb{E}[J_{\boldsymbol{\varphi}_{t+1}}(\boldsymbol{\theta}_{t+1}) - J_{\boldsymbol{\varphi}_t}(\boldsymbol{\theta}_{t+1})] + \mathbb{E}[J_{\boldsymbol{\varphi}_t}(\boldsymbol{\theta}_{t+1})] - J_{\boldsymbol{\varphi}_t}(\boldsymbol{\theta}_t)$$
$$\ge \mathbb{E}[-\left|J_{\boldsymbol{\varphi}_{t+1}}(\boldsymbol{\theta}_{t+1}) - J_{\boldsymbol{\varphi}_t}(\boldsymbol{\theta}_{t+1})\right|] + \mathbb{E}[J_{\boldsymbol{\varphi}_t}(\boldsymbol{\theta}_{t+1})] - J_{\boldsymbol{\varphi}_t}(\boldsymbol{\theta}_t)$$
$$\ge -D_J\mathbb{E}\|\boldsymbol{\varphi}_{t+1} - \boldsymbol{\varphi}_t\|_2 + \mathbb{E}[J_{\boldsymbol{\varphi}_t}(\boldsymbol{\theta}_{t+1})] - J_{\boldsymbol{\varphi}_t}(\boldsymbol{\theta}_t).$$

For simplicity, we denote $\xi = (s,a,s')$ and

$$h_\theta(\boldsymbol{\theta},\boldsymbol{\omega},\boldsymbol{\varphi},\xi) = \left(\tilde{r}_{\boldsymbol{\varphi},\boldsymbol{\theta}}(s,a) + (\gamma\phi(s') - \phi(s))^\top\boldsymbol{\omega}\right)\nabla_{\boldsymbol{\theta}}\log\pi_{\boldsymbol{\theta}}(a|s),$$
$$\bar{h}_\theta(\boldsymbol{\theta},\boldsymbol{\omega},\boldsymbol{\varphi},\nu) = \mathbb{E}_{\nu,\pi_{\boldsymbol{\theta}},\mathcal{P}}[h_\theta(\boldsymbol{\theta},\boldsymbol{\omega},\boldsymbol{\varphi},\xi)],$$

thereby

$$\boldsymbol{\theta}_{t+1} = \boldsymbol{\theta}_t + \eta_t^\theta h_\theta(\boldsymbol{\theta}_t,\boldsymbol{\omega}_t,\boldsymbol{\varphi}_t,\xi_t), \quad \mathbb{E}[\boldsymbol{\theta}_{t+1}|\boldsymbol{\theta}_t] = \boldsymbol{\theta}_t + \eta_t^\theta\bar{h}_\theta(\boldsymbol{\theta}_t,\boldsymbol{\omega}_t,\boldsymbol{\varphi}_t,\hat{\nu}_t).$$

Then we apply the Taylor expansion on $J_{\varphi_t}(\theta)$ around $\theta_t$. The second-order term is propotional to $\eta_t^{\theta 2}$ as the gradient has bounded variance, while the first-order term can be decomposed into three parts, associated with approximation error, critic error and Markovian noise, respectively.

$$
\begin{aligned}
\mathbb{E}[J_{\varphi_t}(\theta_{t+1})] - J_{\varphi_t}(\theta_t) \geq & \mathbb{E}\langle \nabla_\theta J_{\varphi_t}(\theta_t), \theta_{t+1} - \theta_t \rangle - \frac{S_J}{2}\mathbb{E}\|\theta_{t+1} - \theta_t\|_2^2 \\
\geq & \eta_t^\theta \langle \nabla_\theta J_{\varphi_t}(\theta_t), \bar{h}_\theta(\theta_t, \omega_t, \varphi_t, \hat{\nu}_t) \rangle - \frac{S_J \eta_t^{\theta 2}}{2}\mathbb{E}\|h_\theta(\theta_t, \omega_t, \varphi_t, \xi_t)\|_2^2 \\
\geq & \eta_t^\theta \langle \nabla_\theta J_{\varphi_t}(\theta_t), (1-\gamma)\nabla_\theta J_{\varphi_t}(\theta_t) \rangle \\
& + \eta_t^\theta \langle \nabla_\theta J_{\varphi_t}(\theta_t), \bar{h}_\theta(\theta_t, \omega_t^*, \varphi_t, \nu_\rho^{\pi_{\theta_t}}) - (1-\gamma)\nabla_\theta J_{\varphi_t}(\theta_t) \rangle \\
& + \eta_t^\theta \langle \nabla_\theta J_{\varphi_t}(\theta_t), \bar{h}_\theta(\theta_t, \omega_t, \varphi_t, \nu_\rho^{\pi_{\theta_t}}) - \bar{h}_\theta(\theta_t, \omega_t^*, \varphi_t, \nu_\rho^{\pi_{\theta_t}}) \rangle \\
& + \eta_t^\theta \langle \nabla_\theta J_{\varphi_t}(\theta_t), \bar{h}_\theta(\theta_t, \omega_t, \varphi_t, \hat{\nu}_t) - \bar{h}_\theta(\theta_t, \omega_t, \varphi_t, \nu_\rho^{\pi_{\theta_t}}) \rangle \\
& - \frac{L^2 C_\delta^2 S_J \eta_t^{\theta 2}}{2} \\
\geq & \eta_t^\theta (1-\gamma)\|\nabla_\theta J_{\varphi_t}(\theta_t)\|_2^2 \\
& - \eta_t^\theta \|\nabla_\theta J_{\varphi_t}(\theta_t)\|_2 \underbrace{\left\|\bar{h}_\theta(\theta_t, \omega_t^*, \varphi_t, \nu_\rho^{\pi_{\theta_t}}) - (1-\gamma)\nabla_\theta J_{\varphi_t}(\theta_t)\right\|_2}_{I_1} \\
& - \eta_t^\theta \|\nabla_\theta J_{\varphi_t}(\theta_t)\|_2 \underbrace{\left\|\bar{h}_\theta(\theta_t, \omega_t, \varphi_t, \nu_\rho^{\pi_{\theta_t}}) - \bar{h}_\theta(\theta_t, \omega_t^*, \varphi_t, \nu_\rho^{\pi_{\theta_t}})\right\|_2}_{I_2} \\
& - \eta_t^\theta \|\nabla_\theta J_{\varphi_t}(\theta_t)\|_2 \underbrace{\left\|\bar{h}_\theta(\theta_t, \omega_t, \varphi_t, \hat{\nu}_t) - \bar{h}_\theta(\theta_t, \omega_t, \varphi_t, \nu_\rho^{\pi_{\theta_t}})\right\|_2}_{I_3} \\
& - \frac{L^2 C_\delta^2 S_J \eta_t^{\theta 2}}{2}
\end{aligned}
\tag{16}
$$

$I_1$ is associated with the approximation error. Using Proposition 4.3, we have

$$
\begin{aligned}
I_1 = & \left\| \mathbb{E}_{\nu_\rho^{\pi_{\theta_t}}, \pi_{\theta_t}, \mathcal{P}} \left[ \left( \left( \tilde{r}_{\varphi_t, \theta_t}(s, a) + \gamma \widehat{V}_{\omega_t^*}(s') - \widehat{V}_{\omega_t^*}(s) \right) \right. \right. \right. \\
& \left. \left. \left. - \left( \tilde{r}_{\varphi_t, \theta_t}(s, a) + \gamma \widetilde{V}_{\varphi_t}^{\pi_{\theta_t}}(s') - \widetilde{V}_{\varphi_t}^{\pi_{\theta_t}}(s) \right) \right) \nabla_\theta \log \pi_\theta(a|s) \right] \right\|_2 \\
= & \left\| \mathbb{E}_{\nu_\rho^{\pi_{\theta_t}}, \pi_{\theta_t}, \mathcal{P}} \left[ \left( \gamma(\widehat{V}_{\omega_t^*}(s') - \widetilde{V}_{\varphi_t}^{\pi_{\theta_t}}(s')) - (\widehat{V}_{\omega_t^*}(s) - \widetilde{V}_{\varphi_t}^{\pi_{\theta_t}}(s)) \right) \nabla_\theta \log \pi_\theta(a|s) \right] \right\|_2 \\
\leq & L \sqrt{\mathbb{E}_{\nu_\rho^{\pi_{\theta_t}}, \pi_{\theta_t}, \mathcal{P}} \left[ \left( \gamma(\widehat{V}_{\omega^*}(s') - \widetilde{V}_{\varphi_t}^{\pi_{\theta_t}}(s')) - (\widehat{V}_{\omega^*}(s) - \widetilde{V}_{\varphi_t}^{\pi_{\theta_t}}(s)) \right)^2 \right]} \\
\leq & 2\sqrt{2} L \epsilon.
\end{aligned}
$$

$I_2$ is associated with the critic error. We have

$$
\begin{aligned}
I_2 = & \left\| \mathbb{E}_{\nu_\rho^{\pi_{\theta_t}}, \pi_{\theta_t}, \mathcal{P}} \left[ \left( \left( \tilde{r}_{\varphi_t, \theta_t}(s, a) + \gamma \widehat{V}_{\omega_t}(s') - \widehat{V}_{\omega_t}(s) \right) \right. \right. \right. \\
& \left. \left. \left. - \left( \tilde{r}_{\varphi_t, \theta_t}(s, a) + \gamma \widehat{V}_{\omega_t^*}(s') - \widehat{V}_{\omega_t^*}(s) \right) \right) \nabla_\theta \log \pi_\theta(a|s) \right] \right\|_2 \\
= & \left\| \mathbb{E}_{\nu_\rho^{\pi_{\theta_t}}, \pi_{\theta_t}, \mathcal{P}} \left[ (\gamma\phi(s') - \phi(s))^\top (\omega_t - \omega_t^*) \nabla_\theta \log \pi_{\theta_t}(a|s) \right] \right\|_2 \\
\leq & L \sqrt{\mathbb{E}_{\nu_\rho^{\pi_{\theta_t}}, \pi_{\theta_t}, \mathcal{P}} \left[ ((\gamma\phi(s') - \phi(s))^\top (\omega_t - \omega_t^*))^2 \right]} \\
\leq & 2L \|\omega_t - \omega_t^*\|_2.
\end{aligned}
$$

$I_3$ is associated with the Markovian noise. Using Proposition 4.9, we have

$$
\begin{aligned}
I_3 = & \left\| \int_{\mathcal{S}} \mathrm{d}s \left( \hat{\nu}_t(s) - \nu_\rho^{\pi_{\theta_t}}(s) \right) \mathbb{E}_{a, s'} \left[ \hat{\delta}(s, a, s') \nabla_\theta \log \pi_{\theta_t}(a|s) \right] \right\|_2 \\
\leq & L C_\delta \|\hat{\nu}_t - \nu_\rho^{\pi_{\theta_t}}\|_1
\end{aligned}
$$

$$\leq L^2 C_\delta^2 L_\nu \sum_{k=0}^{t-1} \gamma^{t-1-k} \eta_k^\theta + L C_\delta \gamma^t \|\rho - \nu_\rho^{\pi_{\theta_0}}\|_1.$$

Combining the above, we can derive from (16) that

$$
\begin{aligned}
(1-\gamma)\|\nabla_{\boldsymbol{\theta}} J_{\boldsymbol{\varphi}_t}(\boldsymbol{\theta}_t)\|_2^2 &\leq \frac{1}{\eta_t^\theta} \left( \mathbb{E}[J_{\boldsymbol{\varphi}_{t+1}}(\boldsymbol{\theta}_{t+1})] - J_{\boldsymbol{\varphi}_t}(\boldsymbol{\theta}_t) \right) + \frac{D_J \mathbb{E} \|\boldsymbol{\varphi}_{t+1} - \boldsymbol{\varphi}_t\|_2}{\eta_t^\theta} \\
&\quad + \|\nabla_{\boldsymbol{\theta}} J_{\boldsymbol{\varphi}_t}(\boldsymbol{\theta}_t)\|_2 (I_1 + I_2 + I_3) + \frac{L^2 C_\delta^2 S_J \eta_t^\theta}{2} \\
&\leq \frac{1}{\eta_t^\theta} \left( \mathbb{E}[J_{\boldsymbol{\varphi}_{t+1}}(\boldsymbol{\theta}_{t+1})] - J_{\boldsymbol{\varphi}_t}(\boldsymbol{\theta}_t) \right) + \frac{D_J \mathbb{E} \|\boldsymbol{\varphi}_{t+1} - \boldsymbol{\varphi}_t\|_2}{\eta_t^\theta} \\
&\quad + 2\sqrt{2} L L_J \epsilon + 2L \|\boldsymbol{\omega}_t - \boldsymbol{\omega}_t^*\|_2 \|\nabla_{\boldsymbol{\theta}} J_{\boldsymbol{\varphi}_t}(\boldsymbol{\theta}_t)\|_2 \\
&\quad + L^2 C_\delta^2 L_J L_\nu \sum_{k=0}^{t-1} \gamma^{t-1-k} \eta_k^\theta + L C_\delta L_J \gamma^t \|\rho - \nu_\rho^{\pi_{\theta_0}}\|_1 \\
&\quad + \frac{L^2 C_\delta^2 S_J}{2} \eta_t^\theta.
\end{aligned}
$$

Summing over iterations, we have

$$
\begin{aligned}
(1-\gamma) \sum_{t=T/2}^{T-1} \mathbb{E} \|\nabla_{\boldsymbol{\theta}} J_{\boldsymbol{\varphi}_t}(\boldsymbol{\theta}_t)\|_2^2 &\leq \underbrace{\sum_{t=T/2}^{T-1} \frac{1}{\eta_t^\theta} \left( \mathbb{E}[J_{\boldsymbol{\varphi}_{t+1}}(\boldsymbol{\theta}_{t+1})] - \mathbb{E}[J_{\boldsymbol{\varphi}_t}(\boldsymbol{\theta}_t)] \right)}_{S_1} + D_J \underbrace{\sum_{t=T/2}^{T-1} \frac{\mathbb{E} \|\boldsymbol{\varphi}_{t+1} - \boldsymbol{\varphi}_t\|_2}{\eta_t^\theta}}_{S_2} \\
&\quad + \sqrt{2} T L L_J \epsilon + 2L \underbrace{\sum_{t=T/2}^{T-1} \|\boldsymbol{\omega}_t - \boldsymbol{\omega}_t^*\|_2 \|\nabla_{\boldsymbol{\theta}} J_{\boldsymbol{\varphi}_t}(\boldsymbol{\theta}_t)\|_2}_{S_3} \\
&\quad + L^2 C_\delta^2 L_J L_\nu \underbrace{\sum_{t=T/2}^{T-1} \sum_{k=0}^{t-1} \gamma^{t-1-k} \eta_k^\theta}_{S_4} + L C_\delta L_J \|\rho - \nu_\rho^{\pi_{\theta_0}}\|_1 \underbrace{\sum_{j=T/2}^{T-1} \gamma^j}_{S_5} \\
&\quad + \frac{L^2 C_\delta^2 S_J}{2} \underbrace{\sum_{t=T/2}^{T-1} \eta_t^\theta}_{S_6}. \quad\quad (17)
\end{aligned}
$$

For $S_1$, by applying the telescoping skill, we have

$$
\begin{aligned}
S_1 &= \sum_{t=T/2+1}^{T-1} \left( \frac{1}{\eta_{t-1}^\theta} - \frac{1}{\eta_t^\theta} \right) \mathbb{E}[J_{\boldsymbol{\varphi}_t}(\boldsymbol{\theta}_t)] + \frac{\mathbb{E}[J_{\boldsymbol{\varphi}_T}(\boldsymbol{\theta}_T)]}{\eta_{T-1}^\theta} - \frac{\mathbb{E}[J_{\boldsymbol{\varphi}_{T/2}}(\boldsymbol{\theta}_{T/2})]}{\eta_{T/2}^\theta} \\
&\leq \sum_{t=T/2+1}^{T-1} \left( \frac{1}{\eta_t^\theta} - \frac{1}{\eta_{t-1}^\theta} \right) C_J + \frac{C_J}{\eta_{T-1}^\theta} + \frac{C_J}{\eta_{T/2}^\theta} \\
&= \frac{2 C_J}{\eta_{T-1}^\theta} \\
&= O\left( \sqrt{T} \right).
\end{aligned}
$$

For $S_2$, by applying the Cauchy-Schwartz inequality, we have

$$
S_2 \leq \sqrt{\sum_{t=T/2}^{T-1} \mathbb{E} \|\boldsymbol{\varphi}_{t+1} - \boldsymbol{\varphi}_t\|_2^2} \sqrt{\sum_{t=T/2}^{T-1} \frac{1}{\eta_t^{\theta^2}}}
$$

$$= \sqrt{\frac{TF_T}{2} \sum_{t=T/2}^{T-1} \frac{1}{\eta_t^{\theta 2}}}$$

$$= O\left(\sqrt{TF_T}\right),$$

where the last equality is due to the fact that

$$\sum_{t=T/2}^{T-1} \frac{1}{\eta_t^{\theta 2}} = \frac{1}{c_{\boldsymbol{\theta}}^2} \sum_{t=T/2+1}^{T} \frac{1}{t} = \frac{1}{c_{\boldsymbol{\theta}}^2}(H_T - H_{T/2}) \sim \frac{\ln 2}{c_{\boldsymbol{\theta}}^2}.$$

For $S_3$, by applying the Cauchy-Schwartz inequality, we have

$$S_3 \le \sqrt{\sum_{t=T/2}^{T-1} \mathbb{E}\|\boldsymbol{\omega}_t - \boldsymbol{\omega}_t^*\|_2^2} \sqrt{\sum_{t=T/2}^{T-1} \|\nabla_{\boldsymbol{\theta}} J_{\boldsymbol{\varphi}_t}(\boldsymbol{\theta}_t)\|_2^2}$$

$$= \frac{T}{2} \sqrt{\frac{1}{T/2} \sum_{t=T/2}^{T-1} \mathbb{E}\|\boldsymbol{\omega}_t - \boldsymbol{\omega}_t^*\|_2^2} \sqrt{\frac{1}{T/2} \sum_{t=T/2}^{T-1} \|\nabla_{\boldsymbol{\theta}} J_{\boldsymbol{\varphi}_t}(\boldsymbol{\theta}_t)\|_2^2}$$

$$= \frac{T}{2} \sqrt{G_T W_T}.$$

For $S_4$, $S_5$ and $S_6$, we have

$$S_4 \le \sum_{t=0}^{T-1} \sum_{k=0}^{t-1} \gamma^{t-1-k} \eta_k^{\theta} = \sum_{t=0}^{T-1} \eta_t^{\theta} \sum_{j=0}^{T-t-1} \gamma^j \le \sum_{t=0}^{T-1} \frac{\eta_t^{\theta}}{1-\gamma}$$

$$= \frac{1}{c_{\boldsymbol{\theta}}(1-\gamma)} \sum_{t=1}^{T} \frac{1}{\sqrt{t}} = O\left(\sqrt{T}\right),$$

$$S_5 \le \frac{1}{\gamma^{T/2}(1-\gamma)},$$

$$S_6 = \frac{1}{c_{\boldsymbol{\theta}}} \sum_{t=T/2+1}^{T} \frac{1}{\sqrt{t}} = O\left(\sqrt{T}\right).$$

Plug $S_1$, $S_2$, $S_3$, $S_4$, $S_5$ and $S_6$ into (17) and divide both sides by $(1-\gamma)\frac{T}{2}$, we obtain

$$G_T \le \frac{2L}{1-\gamma} \sqrt{G_T W_T} + O\left(\sqrt{\frac{F_T}{T}}\right) + O\left(\frac{1}{\sqrt{T}}\right) + O(\epsilon),$$

thus completes the proof.

$\square$

## C.2 STEP 2: BOUNDING THE CRITIC ERROR

**Theorem C.2 (Critic Update)** *Tate $\eta_t^\theta = \frac{c_\theta}{\sqrt{t}}$, $\eta_t^\omega = \frac{c_\omega}{\sqrt{t}}$ where $c_\theta$ and $c_\omega$ are constants such that $\frac{c_\theta}{c_\omega} \le \frac{\lambda}{LS_\omega}$, then*

$$W_T \le 2(1-\gamma)L_\omega \frac{c_\theta}{c_\omega}\sqrt{W_T G_T} + O\left(F_T\sqrt{T}\right) + O\left(\sqrt{\frac{F_T}{T}}\right) + O\left(\frac{1}{\sqrt{T}}\right) + O(\epsilon).$$

*Proof*    For simplicity, we denote $\xi = (s, a, s')$ and
$$h_\omega(\boldsymbol{\theta}, \boldsymbol{\omega}, \boldsymbol{\varphi}, \xi) = \left(\tilde{r}_{\boldsymbol{\varphi}, \boldsymbol{\theta}}(s, a) + (\gamma\phi(s') - \phi(s))^\top \boldsymbol{\omega}\right)\phi(s),$$
$$\bar{h}_\omega(\boldsymbol{\theta}, \boldsymbol{\omega}, \boldsymbol{\varphi}, \nu) = \mathbb{E}_{\nu, \pi_{\boldsymbol{\theta}}, \mathcal{P}}[h_\omega(\boldsymbol{\theta}, \boldsymbol{\omega}, \boldsymbol{\varphi}, \xi)].$$

According to the critic update rule (9), we have

$$
\begin{aligned}
\left\|\boldsymbol{\omega}_{t+1} - \boldsymbol{\omega}_{t+1}^*\right\|_2^2 &= \left\|\Pi_{C_\omega}\left(\boldsymbol{\omega}_t + \eta_t^\omega h_\omega(\boldsymbol{\theta}_t, \boldsymbol{\omega}_t, \boldsymbol{\varphi}_t, \xi_t)\right) - \Pi_{C_\omega}\left(\boldsymbol{\omega}_{t+1}^*\right)\right\|_2^2 \\
&\le \left\|\boldsymbol{\omega}_t + \eta_t^\omega h_\omega(\boldsymbol{\theta}_t, \boldsymbol{\omega}_t, \boldsymbol{\varphi}_t, \xi_t) - \boldsymbol{\omega}_{t+1}^*\right\|_2^2 \\
&= \left\|(\boldsymbol{\omega}_t - \boldsymbol{\omega}_t^*) + \eta_t^\omega h_\omega(\boldsymbol{\theta}_t, \boldsymbol{\omega}_t, \boldsymbol{\varphi}_t, \xi_t) + (\boldsymbol{\omega}_t^* - \boldsymbol{\omega}_{t+1}^*)\right\|_2^2 \\
&= \left\|\boldsymbol{\omega}_t - \boldsymbol{\omega}_t^*\right\|_2^2 + \left\|\eta_t^\omega h_\omega(\boldsymbol{\theta}_t, \boldsymbol{\omega}_t, \boldsymbol{\varphi}_t, \xi_t) + (\boldsymbol{\omega}_t^* - \boldsymbol{\omega}_{t+1}^*)\right\|_2^2 \\
&\quad + 2\left\langle \boldsymbol{\omega}_t - \boldsymbol{\omega}_t^*, \eta_t^\omega h_\omega(\boldsymbol{\theta}_t, \boldsymbol{\omega}_t, \boldsymbol{\varphi}_t, \xi_t) + (\boldsymbol{\omega}_t^* - \boldsymbol{\omega}_{t+1}^*)\right\rangle \\
&\le \left\|\boldsymbol{\omega}_t - \boldsymbol{\omega}_t^*\right\|_2^2 + 2\eta_t^{\omega 2}\left\|h_\omega(\boldsymbol{\theta}_t, \boldsymbol{\omega}_t, \boldsymbol{\varphi}_t, \xi_t)\right\|_2^2 + 2\left\|\boldsymbol{\omega}_t^* - \boldsymbol{\omega}_{t+1}^*\right\|_2^2 \\
&\quad + 2\eta_t^\omega\left\langle \boldsymbol{\omega}_t - \boldsymbol{\omega}_t^*, h_\omega(\boldsymbol{\theta}_t, \boldsymbol{\omega}_t, \boldsymbol{\varphi}_t, \xi_t)\right\rangle + 2\left\langle \boldsymbol{\omega}_t - \boldsymbol{\omega}_t^*, \boldsymbol{\omega}_t^* - \boldsymbol{\omega}_{t+1}^*\right\rangle. \quad (18)
\end{aligned}
$$

To capture the evolution of the critic error $\|\boldsymbol{\omega}_t - \boldsymbol{\omega}_t^*\|_2^2$, we need to bound the other four terms on the right-hand side of (18). By taking the expectation, the two quadratic terms can be bounded by

$$\mathbb{E}\left\|h_\omega(\boldsymbol{\theta}_t, \boldsymbol{\omega}_t, \boldsymbol{\varphi}_t, \xi_t)\right\|_2^2 = \mathbb{E}_{\hat{\nu}_t, \pi_{\boldsymbol{\theta}_t}, \mathcal{P}}\left\|\hat{\delta}(s_t, a_t, s_t')\phi(s_t)\right\|_2^2 \le C_\delta^2 \quad (19)$$

and

$$
\begin{aligned}
\mathbb{E}\left\|\boldsymbol{\omega}_t^* - \boldsymbol{\omega}_{t+1}^*\right\|_2^2 &\le \mathbb{E}\left[(L_\omega\|\boldsymbol{\theta}_{t+1} - \boldsymbol{\theta}_t\|_2 + D_\omega\|\boldsymbol{\varphi}_{t+1} - \boldsymbol{\varphi}_t\|_2)^2\right] \\
&\le 2L_\omega^2\eta_t^{\theta 2}\mathbb{E}\left\|h_\theta(\boldsymbol{\theta}_t, \boldsymbol{\omega}_t, \boldsymbol{\varphi}_t, \xi_t)\right\|_2^2 + 2D_\omega^2\mathbb{E}\|\boldsymbol{\varphi}_{t+1} - \boldsymbol{\varphi}_t\|_2^2 \\
&\le 2L^2C_\delta^2L_\omega^2\eta_t^{\theta 2} + 2D_\omega^2\mathbb{E}\|\boldsymbol{\varphi}_{t+1} - \boldsymbol{\varphi}_t\|_2^2. \quad (20)
\end{aligned}
$$

The expectation of first inner-product term can be decomposed as the follows:

$$
\begin{aligned}
\mathbb{E}\left\langle \boldsymbol{\omega}_t - \boldsymbol{\omega}_t^*, h_\omega(\boldsymbol{\theta}_t, \boldsymbol{\omega}_t, \boldsymbol{\varphi}_t, \xi_t)\right\rangle &= \left\langle \boldsymbol{\omega}_t - \boldsymbol{\omega}_t^*, \bar{h}_\omega(\boldsymbol{\theta}_t, \boldsymbol{\omega}_t, \boldsymbol{\varphi}_t, \hat{\nu}_t)\right\rangle \\
&= \underbrace{\left\langle \boldsymbol{\omega}_t - \boldsymbol{\omega}_t^*, \bar{h}_\omega(\boldsymbol{\theta}_t, \boldsymbol{\omega}_t^*, \boldsymbol{\varphi}_t, \nu_\rho^{\pi_{\boldsymbol{\theta}_t}})\right\rangle}_{J_1} \\
&\quad + \underbrace{\left\langle \boldsymbol{\omega}_t - \boldsymbol{\omega}_t^*, \bar{h}_\omega(\boldsymbol{\theta}_t, \boldsymbol{\omega}_t, \boldsymbol{\varphi}_t, \nu_\rho^{\pi_{\boldsymbol{\theta}_t}}) - \bar{h}_\omega(\boldsymbol{\theta}_t, \boldsymbol{\omega}_t^*, \boldsymbol{\varphi}_t, \nu_\rho^{\pi_{\boldsymbol{\theta}_t}})\right\rangle}_{J_2} \\
&\quad + \underbrace{\left\langle \boldsymbol{\omega}_t - \boldsymbol{\omega}_t^*, \bar{h}_\omega(\boldsymbol{\theta}_t, \boldsymbol{\omega}_t, \boldsymbol{\varphi}_t, \hat{\nu}_t) - \bar{h}_\omega(\boldsymbol{\theta}_t, \boldsymbol{\omega}_t, \boldsymbol{\varphi}_t, \nu_\rho^{\pi_{\boldsymbol{\theta}_t}})\right\rangle}_{J_3}.
\end{aligned}
$$

According to the definition of $\boldsymbol{\omega}_t^*$, it should be a stationary point of the update rule, hence we have

$$J_1 = 0.$$

$J_2$ is associated with the critic error. We have

$$
\begin{aligned}
J_2 &= \left\langle \boldsymbol{\omega}_t - \boldsymbol{\omega}_t^*, \mathbb{E}_{\nu_\rho^{\pi_{\boldsymbol{\theta}_t}}, \pi_{\boldsymbol{\theta}}, \mathcal{P}}[(\gamma\phi(s') - \phi(s))^\top(\boldsymbol{\omega}_t - \boldsymbol{\omega}_t^*)\phi(s)]\right\rangle \\
&= \left\langle \boldsymbol{\omega}_t - \boldsymbol{\omega}_t^*, -A_{\boldsymbol{\theta}_t}(\boldsymbol{\omega}_t - \boldsymbol{\omega}_t^*)\right\rangle \\
&\le -\lambda\left\|\boldsymbol{\omega}_t - \boldsymbol{\omega}_t^*\right\|_2^2.
\end{aligned}
$$

$J_3$ is associated with the Markovian noise. Using Lemma 4.9, we have

$$J_3 \leq \|\boldsymbol{\omega}_t - \boldsymbol{\omega}_t^*\|_2 \left\| \bar{h}_\omega(\boldsymbol{\theta}_t, \boldsymbol{\omega}_t, \boldsymbol{\varphi}_t, \hat{\nu}_t) - \bar{h}_\omega(\boldsymbol{\theta}_t, \boldsymbol{\omega}_t, \boldsymbol{\varphi}_t, \nu_\rho^{\pi_{\boldsymbol{\theta}_t}}) \right\|_2$$

$$\leq 2C_{\boldsymbol{\omega}} \left\| \int_{\mathcal{S}} \mathrm{d}s \left( \hat{\nu}_t(s) - \nu_\rho^{\pi_{\boldsymbol{\theta}_t}}(s) \right) \mathbb{E}_{a \sim \pi_{\boldsymbol{\theta}_t}(\cdot|s), s' \sim \mathcal{P}(\cdot|s,a)} \left[ \hat{\delta}(s, a, s') \phi(s) \right] \right\|_2$$

$$\leq 2C_{\boldsymbol{\omega}} C_\delta \| \hat{\nu}_t - \nu_\rho^{\pi_{\boldsymbol{\theta}_t}} \|_1$$

$$\leq 2LC_{\boldsymbol{\omega}} C_\delta^2 L_\nu \sum_{k=0}^{t-1} \gamma^{t-1-k} \eta_k^\theta + 2C_{\boldsymbol{\omega}} C_\delta \gamma^t \| \rho - \nu_\rho^{\pi_{\boldsymbol{\theta}_0}} \|_1.$$

Hence, we have

$$\mathbb{E} \langle \boldsymbol{\omega}_t - \boldsymbol{\omega}_t^*, h_\omega(\boldsymbol{\theta}_t, \boldsymbol{\omega}_t, \boldsymbol{\varphi}_t, \xi_t) \rangle \leq -\lambda \|\boldsymbol{\omega}_t - \boldsymbol{\omega}_t^*\|_2^2 + 2LC_{\boldsymbol{\omega}} C_\delta^2 L_\nu \sum_{k=0}^{t-1} \gamma^{t-1-k} \eta_k^\theta + 2C_{\boldsymbol{\omega}} C_\delta \gamma^t \| \rho - \nu_\rho^{\pi_{\boldsymbol{\theta}_0}} \|_1.$$

$$(21)$$

The analysis of the last term in (18) is similar to the analysis of the actor error. We first leverage the Lipschitz continuity of $\boldsymbol{\omega}^*(\boldsymbol{\varphi}, \boldsymbol{\theta})$ with respect to $\boldsymbol{\varphi}$, then apply the Taylor expansion of $\boldsymbol{\omega}^*(\boldsymbol{\varphi}, \boldsymbol{\theta})$ with respect to $\boldsymbol{\theta}$ around $\boldsymbol{\theta}_t$:

$$\mathbb{E} \langle \boldsymbol{\omega}_t - \boldsymbol{\omega}_t^*, \boldsymbol{\omega}_t^* - \boldsymbol{\omega}_{t+1}^* \rangle = \mathbb{E} \langle \boldsymbol{\omega}_t - \boldsymbol{\omega}_t^*, \boldsymbol{\omega}^*(\boldsymbol{\varphi}_t, \boldsymbol{\theta}_t) - \boldsymbol{\omega}^*(\boldsymbol{\varphi}_{t+1}, \boldsymbol{\theta}_t) \rangle$$

$$+ \mathbb{E} \langle \boldsymbol{\omega}_t - \boldsymbol{\omega}_t^*, \boldsymbol{\omega}^*(\boldsymbol{\varphi}_{t+1}, \boldsymbol{\theta}_t) - \boldsymbol{\omega}^*(\boldsymbol{\varphi}_{t+1}, \boldsymbol{\theta}_{t+1}) \rangle$$

$$= \mathbb{E} \langle \boldsymbol{\omega}_t - \boldsymbol{\omega}_t^*, \boldsymbol{\omega}^*(\boldsymbol{\varphi}_t, \boldsymbol{\theta}_t) - \boldsymbol{\omega}^*(\boldsymbol{\varphi}_{t+1}, \boldsymbol{\theta}_t) \rangle$$

$$+ \mathbb{E} \langle \boldsymbol{\omega}_t - \boldsymbol{\omega}_t^*, \boldsymbol{\omega}^*(\boldsymbol{\varphi}_t, \boldsymbol{\theta}_t) - \boldsymbol{\omega}^*(\boldsymbol{\varphi}_{t+1}, \boldsymbol{\theta}_t) - \nabla_{\boldsymbol{\theta}} \boldsymbol{\omega}^*(\boldsymbol{\varphi}_{t+1}, \boldsymbol{\theta}_t)^\top (\boldsymbol{\theta}_t - \boldsymbol{\theta}_{t+1}) \rangle$$

$$+ \mathbb{E} \langle \boldsymbol{\omega}_t - \boldsymbol{\omega}_t^*, \nabla_{\boldsymbol{\theta}} \boldsymbol{\omega}^*(\boldsymbol{\varphi}_{t+1}, \boldsymbol{\theta}_t)^\top (\boldsymbol{\theta}_t - \boldsymbol{\theta}_{t+1}) \rangle$$

$$\leq D_{\boldsymbol{\omega}} \|\boldsymbol{\omega}_t - \boldsymbol{\omega}_t^*\|_2 \mathbb{E} \|\boldsymbol{\varphi}_{t+1} - \boldsymbol{\varphi}_t\|_2 + \frac{S_{\boldsymbol{\omega}}}{2} \|\boldsymbol{\omega}_t - \boldsymbol{\omega}_t^*\|_2 \mathbb{E} \|\boldsymbol{\theta}_{t+1} - \boldsymbol{\theta}_t\|_2^2$$

$$+ \mathbb{E} \langle \boldsymbol{\omega}_t - \boldsymbol{\omega}_t^*, \nabla_{\boldsymbol{\theta}} \boldsymbol{\omega}^*(\boldsymbol{\varphi}_{t+1}, \boldsymbol{\theta}_t)^\top (\boldsymbol{\theta}_t - \boldsymbol{\theta}_{t+1}) \rangle$$

$$\leq L^2 C_\delta^2 C_{\boldsymbol{\omega}} S_{\boldsymbol{\omega}} \eta_t^{\theta^2} + 2C_{\boldsymbol{\omega}} D_{\boldsymbol{\omega}} \mathbb{E} \|\boldsymbol{\varphi}_{t+1} - \boldsymbol{\varphi}_t\|_2$$

$$+ \underbrace{\mathbb{E} \langle \boldsymbol{\omega}_t - \boldsymbol{\omega}_t^*, \nabla_{\boldsymbol{\theta}} \boldsymbol{\omega}^*(\boldsymbol{\varphi}_{t+1}, \boldsymbol{\theta}_t)^\top (\boldsymbol{\theta}_t - \boldsymbol{\theta}_{t+1}) \rangle}_{I},$$

where the last inequality uses the facts that

$$\mathbb{E} \|\boldsymbol{\theta}_t - \boldsymbol{\theta}_{t+1}\|_2^2 \leq L^2 C_\delta^2 \eta_t^{\theta^2} \quad \text{and} \quad \|\boldsymbol{\omega}_t - \boldsymbol{\omega}_t^*\|_2^2 \leq 2C_{\boldsymbol{\omega}}.$$

The remaining inner product term $I$ can then be decomposed into terms related to $I_1$, $I_2$ and $I_3$.

$$I = -\eta_t^\theta \langle \boldsymbol{\omega}_t - \boldsymbol{\omega}_t^*, \nabla_{\boldsymbol{\theta}} \boldsymbol{\omega}^*(\boldsymbol{\varphi}_{t+1}, \boldsymbol{\theta}_t)^\top \bar{h}_\theta(\boldsymbol{\theta}_t, \boldsymbol{\omega}_t, \boldsymbol{\varphi}_t, \xi_t) \rangle$$

$$= -(1-\gamma)\eta_t^\theta \langle \boldsymbol{\omega}_t - \boldsymbol{\omega}_t^*, \nabla_{\boldsymbol{\theta}} \boldsymbol{\omega}^*(\boldsymbol{\varphi}_{t+1}, \boldsymbol{\theta}_t)^\top \nabla_{\boldsymbol{\theta}} J_{\boldsymbol{\varphi}_t}(\boldsymbol{\theta}_t) \rangle$$

$$- \eta_t^\theta \langle \boldsymbol{\omega}_t - \boldsymbol{\omega}_t^*, \nabla_{\boldsymbol{\theta}} \boldsymbol{\omega}^*(\boldsymbol{\varphi}_{t+1}, \boldsymbol{\theta}_t)^\top \left( \bar{h}_\theta(\boldsymbol{\theta}_t, \boldsymbol{\omega}_t^*, \boldsymbol{\varphi}_t, \nu_\rho^{\pi_{\boldsymbol{\theta}_t}}) - (1-\gamma) \nabla_{\boldsymbol{\theta}} J_{\boldsymbol{\varphi}_t}(\boldsymbol{\theta}_t) \right) \rangle$$

$$- \eta_t^\theta \langle \boldsymbol{\omega}_t - \boldsymbol{\omega}_t^*, \nabla_{\boldsymbol{\theta}} \boldsymbol{\omega}^*(\boldsymbol{\varphi}_{t+1}, \boldsymbol{\theta}_t)^\top \left( \bar{h}_\theta(\boldsymbol{\theta}_t, \boldsymbol{\omega}_t, \boldsymbol{\varphi}_t, \nu_\rho^{\pi_{\boldsymbol{\theta}_t}}) - \bar{h}_\theta(\boldsymbol{\theta}_t, \boldsymbol{\omega}_t^*, \boldsymbol{\varphi}_t, \nu_\rho^{\pi_{\boldsymbol{\theta}_t}}) \right) \rangle$$

$$- \eta_t^\theta \langle \boldsymbol{\omega}_t - \boldsymbol{\omega}_t^*, \nabla_{\boldsymbol{\theta}} \boldsymbol{\omega}^*(\boldsymbol{\varphi}_{t+1}, \boldsymbol{\theta}_t)^\top \left( \bar{h}_\theta(\boldsymbol{\theta}_t, \boldsymbol{\omega}_t, \boldsymbol{\varphi}_t, \hat{\nu}_t) - \bar{h}_\theta(\boldsymbol{\theta}_t, \boldsymbol{\omega}_t, \boldsymbol{\varphi}_t, \nu_\rho^{\pi_{\boldsymbol{\theta}_t}}) \right) \rangle$$

$$\leq L^2 C_\delta^2 C_{\boldsymbol{\omega}} S_{\boldsymbol{\omega}} \eta_t^{\theta^2} + 2C_{\boldsymbol{\omega}} D_{\boldsymbol{\omega}} \mathbb{E} \|\boldsymbol{\varphi}_{t+1} - \boldsymbol{\varphi}_t\|_2$$

$$+ (1-\gamma) S_{\boldsymbol{\omega}} \eta_t^\theta \|\boldsymbol{\omega}_t - \boldsymbol{\omega}_t^*\|_2 \|\nabla_{\boldsymbol{\theta}} J_{\boldsymbol{\varphi}_t}(\boldsymbol{\theta}_t)\|_2$$

$$+ S_{\boldsymbol{\omega}} \eta_t^\theta \|\boldsymbol{\omega}_t - \boldsymbol{\omega}_t^*\|_2 \underbrace{\left\| \bar{h}_\theta(\boldsymbol{\theta}_t, \boldsymbol{\omega}_t^*, \boldsymbol{\varphi}_t, \nu_\rho^{\pi_{\boldsymbol{\theta}_t}}) - (1-\gamma) \nabla_{\boldsymbol{\theta}} J_{\boldsymbol{\varphi}_t}(\boldsymbol{\theta}_t) \right\|}_{I_1}$$

$$+ S_{\boldsymbol{\omega}} \eta_t^\theta \|\boldsymbol{\omega}_t - \boldsymbol{\omega}_t^*\|_2 \underbrace{\left\| \bar{h}_\theta(\boldsymbol{\theta}_t, \boldsymbol{\omega}_t, \boldsymbol{\varphi}_t, \nu_\rho^{\pi_{\boldsymbol{\theta}_t}}) - \bar{h}_\theta(\boldsymbol{\theta}_t, \boldsymbol{\omega}_t^*, \boldsymbol{\varphi}_t, \nu_\rho^{\pi_{\boldsymbol{\theta}_t}}) \right\|}_{I_2}$$

$$+ S_{\boldsymbol{\omega}} \eta_t^\theta \|\boldsymbol{\omega}_t - \boldsymbol{\omega}_t^*\|_2 \underbrace{\left\| \bar{h}_\theta(\boldsymbol{\theta}_t, \boldsymbol{\omega}_t, \boldsymbol{\varphi}_t, \hat{\nu}_t) - \bar{h}_\theta(\boldsymbol{\theta}_t, \boldsymbol{\omega}_t, \boldsymbol{\varphi}_t, \nu_\rho^{\pi_{\boldsymbol{\theta}_t}}) \right\|}_{I_3},$$

where

$$I_1 \leq 2\sqrt{2}L\epsilon, \quad I_2 \leq 2L\|\boldsymbol{\omega}_t - \boldsymbol{\omega}_t^*\|_2,$$

$$I_3 \leq L^2 C_\delta^2 L_\nu \sum_{k=0}^{t-1} \gamma^{t-1-k}\eta_k^\theta + LC_\delta\gamma^t\|\rho - \nu_\rho^{\pi_{\boldsymbol{\theta}_0}}\|_1.$$

Hence, we have

$$
\begin{aligned}
\left\langle \boldsymbol{\omega}_t - \boldsymbol{\omega}_t^*, \boldsymbol{\omega}_t^* - \boldsymbol{\omega}_{t+1}^* \right\rangle \leq & L^2 C_\delta^2 C_{\boldsymbol{\omega}} S_{\boldsymbol{\omega}} \eta_t^{\theta\,2} + 2C_{\boldsymbol{\omega}} D_{\boldsymbol{\omega}} \|\boldsymbol{\varphi}_{t+1} - \boldsymbol{\varphi}_t\|_2 \\
& + (1-\gamma) S_{\boldsymbol{\omega}} \eta_t^\theta \|\boldsymbol{\omega}_t - \boldsymbol{\omega}_t^*\|_2 \|\nabla_{\boldsymbol{\theta}} J_{\boldsymbol{\varphi}_t}(\boldsymbol{\theta}_t)\|_2 \\
& + 4\sqrt{2}LC_{\boldsymbol{\omega}} S_{\boldsymbol{\omega}} \epsilon \eta_t^\theta + 2LS_{\boldsymbol{\omega}} \eta_t^\theta \|\boldsymbol{\omega}_t - \boldsymbol{\omega}_t^*\|_2^2 \\
& + 2L^2 C_{\boldsymbol{\omega}} C_\delta^2 L_\nu S_{\boldsymbol{\omega}} \eta_t^\theta \sum_{k=0}^{t-1} \gamma^{t-1-k}\eta_k^\theta + 2LC_{\boldsymbol{\omega}} C_\delta S_{\boldsymbol{\omega}} \eta_t^\theta \gamma^t \|\rho - \nu_\rho^{\pi_{\boldsymbol{\theta}_0}}\|_1.
\end{aligned}
$$
(22)

Plugging (19), (20), (21) and (22) into (18) gives

$$
\begin{aligned}
\mathbb{E}\left\|\boldsymbol{\omega}_{t+1} - \boldsymbol{\omega}_{t+1}^*\right\|_2^2 \leq & \left(1 - 2\lambda\eta_t^\omega + 4LS_{\boldsymbol{\omega}}\eta_t^\theta\right) \|\boldsymbol{\omega}_t - \boldsymbol{\omega}_t^*\|_2^2 \\
& + 2(1-\gamma) S_{\boldsymbol{\omega}} \eta_t^\theta \|\boldsymbol{\omega}_t - \boldsymbol{\omega}_t^*\|_2 \|\nabla_{\boldsymbol{\theta}} J_t(\boldsymbol{\theta}_t)\|_2 \\
& + 2D_{\boldsymbol{\omega}}^2 \mathbb{E}\|\boldsymbol{\varphi}_{t+1} - \boldsymbol{\varphi}_t\|_2^2 + 4C_{\boldsymbol{\omega}} D_{\boldsymbol{\omega}} \mathbb{E}\|\boldsymbol{\varphi}_{t+1} - \boldsymbol{\varphi}_t\|_2 \\
& + 4LC_{\boldsymbol{\omega}} C_\delta^2 L_\nu (LS_{\boldsymbol{\omega}}\eta_t^\theta + \eta_t^\omega) \sum_{k=0}^{t-1} \gamma^{t-1-k}\eta_k^\theta \\
& + 4C_{\boldsymbol{\omega}} C_\delta (LS_{\boldsymbol{\omega}}\eta_t^\theta + \eta_t^\omega)\gamma^t \|\rho - \nu_\rho^{\pi_{\boldsymbol{\theta}_0}}\|_1 \\
& + 4L^2 C_\delta^2 L_{\boldsymbol{\omega}}^2 \eta_t^{\theta\,2} + 8\sqrt{2}LC_{\boldsymbol{\omega}} S_{\boldsymbol{\omega}} \epsilon \eta_t^\theta.
\end{aligned}
$$

Note that $\frac{\eta_t^\theta}{\eta_t^\omega} = \frac{c_{\boldsymbol{\theta}}}{c_{\boldsymbol{\omega}}} \leq \frac{\lambda}{LS_{\boldsymbol{\omega}}}$, thus

$$
\begin{aligned}
\lambda \sum_{t=T/2}^{T-1} \mathbb{E}\|\boldsymbol{\omega}_t - \boldsymbol{\omega}_t^*\|_2^2 \leq & \underbrace{\sum_{t=T/2}^{T-1} \frac{1}{\eta_t^\omega} \left( \mathbb{E}\|\boldsymbol{\omega}_t - \boldsymbol{\omega}_t^*\|_2^2 - \mathbb{E}\left\|\boldsymbol{\omega}_{t+1} - \boldsymbol{\omega}_{t+1}^*\right\|_2^2 \right)}_{S_1} \\
& + \underbrace{2(1-\gamma)L_{\boldsymbol{\omega}} \frac{c_{\boldsymbol{\theta}}}{c_{\boldsymbol{\omega}}} \sum_{t=T/2}^{T-1} \|\boldsymbol{\omega}_t - \boldsymbol{\omega}_t^*\|_2 \|\nabla_{\boldsymbol{\theta}} J_t(\boldsymbol{\theta}_t)\|_2}_{S_2} \\
& + \underbrace{2D_{\boldsymbol{\omega}}^2 \sum_{t=T/2}^{T-1} \frac{\mathbb{E}\|\boldsymbol{\varphi}_{t+1} - \boldsymbol{\varphi}_t\|_2^2}{\eta_t^\omega}}_{S_3} + \underbrace{2C_{\boldsymbol{\omega}} D_{\boldsymbol{\omega}} \sum_{t=T/2}^{T-1} \frac{\mathbb{E}\|\boldsymbol{\varphi}_{t+1} - \boldsymbol{\varphi}_t\|_2}{\eta_t^\omega}}_{S_4} \\
& + \underbrace{(1+\lambda)LC_{\boldsymbol{\omega}} C_\delta^2 L_\nu \sum_{t=T/2}^{T-1}\sum_{k=0}^{t-1} \gamma^{t-1-k}\eta_k^\theta}_{S_5} \\
& + \underbrace{(1+\lambda)C_{\boldsymbol{\omega}} C_\delta L_J \|\rho - \nu_\rho^{\pi_{\boldsymbol{\theta}_0}}\|_1 \sum_{j=T/2}^{T-1} \gamma^j}_{S_6} \\
& + \underbrace{4L^2 C_\delta^2 L_{\boldsymbol{\omega}}^2 \frac{c_{\boldsymbol{\theta}}}{c_{\boldsymbol{\omega}}} \sum_{t=T/2}^{T-1} \eta_t^\theta}_{S_7} + 8\sqrt{2}LC_{\boldsymbol{\omega}} S_{\boldsymbol{\omega}} \frac{c_{\boldsymbol{\theta}}}{c_{\boldsymbol{\omega}}}\epsilon
\end{aligned}
$$
(23)

For $S_1$, by applying the telescoping skill, we have

$$S_1 = \sum_{t=T/2+1}^{T-1} \left( \frac{1}{\eta_t^\omega} - \frac{1}{\eta_{t-1}^\omega} \right) \mathbb{E}\|\boldsymbol{\omega}_t - \boldsymbol{\omega}_t^*\|_2^2 - \frac{\mathbb{E}\|\boldsymbol{\omega}_T - \boldsymbol{\omega}_T^*\|_2^2}{\eta_{T-1}^\omega} + \frac{\mathbb{E}\|\boldsymbol{\omega}_{T/2} - \boldsymbol{\omega}_{T/2}^*\|_2^2}{\eta_{T/2}^\omega}$$

$$\leq \sum_{t=T/2+1}^{T-1} \left( \frac{1}{\eta_t^\omega} - \frac{1}{\eta_{t-1}^\omega} \right) 2C_{\boldsymbol{\omega}} + \frac{2C_{\boldsymbol{\omega}}}{\eta_{T-1}^\omega} + \frac{2C_{\boldsymbol{\omega}}}{\eta_{T/2}^\omega}$$

$$= \frac{4C_{\boldsymbol{\omega}}}{\eta_{T-1}^\omega}$$

$$= O\left(\sqrt{T}\right).$$

For $S_2$, by applying the Cauchy-Schwartz inequality, we have

$$S_2 \leq \sqrt{\sum_{t=T/2}^{T-1} \mathbb{E}\|\boldsymbol{\omega}_t - \boldsymbol{\omega}_t^*\|_2^2} \sqrt{\sum_{t=T/2}^{T-1} \|\nabla_{\boldsymbol{\theta}} J_{\boldsymbol{\varphi}_t}(\boldsymbol{\theta}_t)\|_2^2}$$

$$= \frac{T}{2} \sqrt{\frac{1}{T/2} \sum_{t=T/2}^{T-1} \mathbb{E}\|\boldsymbol{\omega}_t - \boldsymbol{\omega}_t^*\|_2^2} \sqrt{\frac{1}{T/2} \sum_{t=T/2}^{T-1} \|\nabla_{\boldsymbol{\theta}} J_{\boldsymbol{\varphi}_t}(\boldsymbol{\theta}_t)\|_2^2}$$

$$= \frac{T}{2} \sqrt{W_T G_T}.$$

For $S_4$, note that $\eta_t^\omega$ is decreasing as $t$ grows, we have

$$S_4 = \sum_{t=T/2}^{T-1} \frac{\mathbb{E}\|\boldsymbol{\varphi}_{t+1} - \boldsymbol{\varphi}_t\|_2^2}{\eta_t^\omega} \leq \frac{1}{\eta_{T-1}^\omega} \sum_{t=T/2}^{T-1} \mathbb{E}\|\boldsymbol{\varphi}_{t+1} - \boldsymbol{\varphi}_t\|_2^2 = O\left(F_T T \sqrt{T}\right)$$

For $S_5$, by applying the Cauchy-Schwartz inequality, we have

$$S_2 \leq \sqrt{\sum_{t=T/2}^{T-1} \mathbb{E}\|\boldsymbol{\varphi}_{t+1} - \boldsymbol{\varphi}_t\|_2^2} \sqrt{\sum_{t=T/2}^{T-1} \frac{1}{\eta_t^{\omega 2}}}$$

$$= \sqrt{\frac{T F_T}{2} \sum_{t=T/2}^{T-1} \frac{1}{\eta_t^{\omega 2}}}$$

$$= O\left(\sqrt{T F_T}\right),$$

where the last equality is due to the fact that

$$\sum_{t=T/2}^{T-1} \frac{1}{\eta_t^{\omega 2}} = \frac{1}{c_{\boldsymbol{\omega}}^2} \sum_{t=T/2+1}^{T} \frac{1}{t} = \frac{1}{c_{\boldsymbol{\omega}}^2} (H_T - H_{T/2}) \sim \frac{\ln 2}{c_{\boldsymbol{\omega}}^2}.$$

For $S_5$, $S_6$ and $S_7$, we have

$$S_5 \leq \sum_{t=0}^{T-1} \sum_{k=0}^{t-1} \gamma^{t-1-k} \eta_k^\theta = \sum_{t=0}^{T-1} \eta_t^\theta \sum_{j=0}^{T-t-1} \gamma^j \leq \sum_{t=0}^{T-1} \frac{\eta_t^\theta}{1-\gamma}$$

$$= \frac{1}{c_{\boldsymbol{\theta}}(1-\gamma)} \sum_{t=1}^{T} \frac{1}{\sqrt{t}} = O\left(\sqrt{T}\right),$$

$$S_6 \leq \frac{1}{\gamma^{T/2}(1-\gamma)},$$

$$S_7 = \frac{1}{c_{\boldsymbol{\theta}}} \sum_{t=T/2+1}^{T} \frac{1}{\sqrt{t}} = O\left(\sqrt{T}\right).$$

Plug $S_1$, $S_2$, $S_3$, $S_4$, $S_5$, $S_6$ and $S_7$ into (23) and divide both sides by $\frac{\lambda T}{2}$, we obtain

$$W_T \leq 2(1-\gamma)L_{\boldsymbol{\omega}}\frac{c_{\boldsymbol{\theta}}}{c_{\boldsymbol{\omega}}}\sqrt{W_T G_T} + O\left(F_T\sqrt{T}\right) + O\left(\sqrt{\frac{F_T}{T}}\right) + O\left(\frac{1}{\sqrt{T}}\right) + O(\epsilon),$$

thus completes the proof.

$\square$

### C.3 STEP 3: SOLVING THE SYSTEM OF INEQUALITIES

**Proof of Theorem 4.7**  According to Theorem C.1 and Theorem C.2, we have

$$(1-\gamma)G_T \leq 2L\sqrt{G_T W_T} + O\left(\sqrt{\frac{F_T}{T}}\right) + O\left(\frac{1}{\sqrt{T}}\right) + O\left(\epsilon\right), \tag{24}$$

$$\frac{1}{1-\gamma}W_T \leq 2L_{\boldsymbol{\omega}}\frac{c_{\boldsymbol{\theta}}}{c_{\boldsymbol{\omega}}}\sqrt{G_T W_T} + O\left(\sqrt{T}F_T\right) + O\left(\frac{F_T}{T}\right) + O\left(\frac{1}{\sqrt{T}}\right) + O(\epsilon). \tag{25}$$

Note that

$$2\sqrt{G_T W_T} = 2\sqrt{\frac{1-\gamma}{2L}G_T \cdot \frac{2L}{1-\gamma}W_T} \leq \frac{1-\gamma}{2L}G_T + \frac{2L}{1-\gamma}W_T. \tag{26}$$

Plug (26) into (24), we have

$$\frac{1-\gamma}{2L}G_T \leq \frac{2L}{1-\gamma}W_T + O\left(\sqrt{\frac{F_T}{T}}\right) + O\left(\frac{1}{\sqrt{T}}\right) + O\left(\epsilon\right). \tag{27}$$

Combining (26) and (27), we have

$$2\sqrt{G_T W_T} \leq \frac{4L}{1-\gamma}W_T + O\left(\sqrt{\frac{F_T}{T}}\right) + O\left(\frac{1}{\sqrt{T}}\right) + O\left(\epsilon\right). \tag{28}$$

Plug (28) into (25), we have

$$\frac{1-8LL_{\boldsymbol{\omega}}\frac{c_{\boldsymbol{\theta}}}{c_{\boldsymbol{\omega}}}}{1-\gamma}W_T \leq O\left(\sqrt{T}F_T\right) + O\left(\frac{F_T}{T}\right) + O\left(\frac{1}{\sqrt{T}}\right) + O(\epsilon).$$

Therefore, when $\frac{c_{\boldsymbol{\theta}}}{c_{\boldsymbol{\omega}}} \leq \frac{1}{16LL_{\boldsymbol{\omega}}}$,

$$W_T = O\left(\frac{1}{\sqrt{T}}\right) + O\left(\sqrt{T}F_T\right) + O\left(\frac{F_T}{T}\right) + O(\epsilon).$$

Combined with (27), we have

$$G_T = O\left(\frac{1}{\sqrt{T}}\right) + O\left(\sqrt{T}F_T\right) + O\left(\frac{F_T}{T}\right) + O(\epsilon).$$

## D PROOF OF PROPOSITIONS, PRELIMINARY LEMMAS AND COROLLARIES

**Proof of Proposition 4.2** For any vector $x$,

$$
\begin{aligned}
x^\top A_{\boldsymbol{\theta}} x &= x^\top \mathbb{E}_{\nu_\rho^{\pi_{\boldsymbol{\theta}}}, \pi_{\boldsymbol{\theta}}, \mathcal{P}} \left[ \phi(s) \left( \phi(s) - \gamma \phi(s') \right)^\top \right] x \\
&= x^\top \left( \mathbb{E}_s \left[ \phi(s)\phi(s)^\top \right] - \gamma \mathbb{E}_{s,s'} \left[ \phi(s)\phi(s')^\top \right] \right) x \\
&= \mathbb{E}_s \left[ x^\top \phi(s)\phi(s)^\top x \right] - \gamma \mathbb{E}_{s,s'} \left[ x^\top \phi(s)\phi(s')^\top x \right]
\end{aligned}
$$

According to the Cauchy-Schwartz inequality,

$$
\mathbb{E}_{s,s'} \left[ x^\top \phi(s)\phi(s')^\top x \right] \leq \sqrt{\mathbb{E}_s \left[ x^\top \phi(s)\phi(s)^\top x \right]} \sqrt{\mathbb{E}_{s'} \left[ x^\top \phi(s')\phi(s')^\top x \right]}.
$$

Note that

$$
\Pr(s' = x) = \frac{\nu_\rho^{\pi_{\boldsymbol{\theta}}}(x) - (1-\gamma)\rho(x)}{\gamma} \leq \frac{\nu_\rho^{\pi_{\boldsymbol{\theta}}}(x)}{\gamma},
$$

so

$$
\mathbb{E}_{s'} \left[ x^\top \phi(s')\phi(s')^\top x \right] \leq \frac{1}{\gamma} \mathbb{E}_s \left[ x^\top \phi(s)\phi(s)^\top x \right],
$$

and hence

$$
\begin{aligned}
x^\top A_{\boldsymbol{\theta}} x &\geq \mathbb{E}_s \left[ x^\top \phi(s)\phi(s)^\top x \right] - \gamma \sqrt{\mathbb{E}_s \left[ x^\top \phi(s)\phi(s)^\top x \right]} \sqrt{\frac{1}{\gamma} \mathbb{E}_s \left[ x^\top \phi(s)\phi(s)^\top x \right]} \\
&= (1 - \sqrt{\gamma}) x^\top \Sigma_\theta x \\
&\geq (1 - \sqrt{\gamma}) \lambda_\Sigma \|x\|_2^2
\end{aligned}
$$

Therefore, $A_{\boldsymbol{\theta}}$ is positive definite with singular values lower-bounded by $\lambda = (1 - \sqrt{\gamma})\lambda_\Sigma$.

**Proof of Proposition 4.3**

$$
\begin{aligned}
LHS &= \sqrt{\mathbb{E}_{\nu_\rho^{\pi_{\boldsymbol{\theta}}}, \pi_{\boldsymbol{\theta}}, \mathcal{P}} \left[ \left( \left( \gamma \widehat{V}_{\boldsymbol{\omega}^*}(s') - \widehat{V}_{\boldsymbol{\omega}^*}(s) \right) - \left( \gamma \widetilde{V}^{\pi_{\boldsymbol{\theta}}}(s') - \widetilde{V}^{\pi_{\boldsymbol{\theta}}}(s) \right) \right)^2 \right]} \\
&\leq \sqrt{\mathbb{E}_{\nu_\rho^{\pi_{\boldsymbol{\theta}}}, \pi_{\boldsymbol{\theta}}, \mathcal{P}} \left[ 2 \left( \gamma \left( \widehat{V}_{\boldsymbol{\omega}^*}(s') - \widetilde{V}^{\pi_{\boldsymbol{\theta}}}(s') \right) \right)^2 + 2 \left( \widehat{V}_{\boldsymbol{\omega}^*}(s) - \widetilde{V}^{\pi_{\boldsymbol{\theta}}}(s) \right)^2 \right]} \\
&\leq \sqrt{2 \mathbb{E}_s \left[ \left( \widehat{V}_{\boldsymbol{\omega}^*}(s) - \widetilde{V}^{\pi_{\boldsymbol{\theta}}}(s) \right)^2 \right] + 2\gamma^2 \mathbb{E}_{s'} \left[ \left( \widehat{V}_{\boldsymbol{\omega}^*}(s') - \widetilde{V}^{\pi_{\boldsymbol{\theta}}}(s') \right)^2 \right]} \\
&\leq \sqrt{2} \left( \underbrace{\sqrt{\mathbb{E}_s \left[ \left( \widehat{V}_{\boldsymbol{\omega}^*}(s) - \widetilde{V}^{\pi_{\boldsymbol{\theta}}}(s) \right)^2 \right]}}_{I_1} + \gamma \underbrace{\sqrt{\mathbb{E}_{s'} \left[ \left( \widehat{V}_{\boldsymbol{\omega}^*}(s') - \widetilde{V}^{\pi_{\boldsymbol{\theta}}}(s') \right)^2 \right]}}_{I_2} \right)
\end{aligned}
$$

According to the definition of $\epsilon$ (13), $I_1 \leq \epsilon$.

For $I_2$, note that

$$
\Pr(s' = x) = \frac{\nu_\rho^{\pi_{\boldsymbol{\theta}}}(x) - (1-\gamma)\rho(x)}{\gamma} \leq \frac{\nu_\rho^{\pi_{\boldsymbol{\theta}}}(x)}{\gamma},
$$

so

$$
I_2 \leq \gamma \sqrt{\frac{1}{\gamma} \mathbb{E}_s \left[ \left( \widehat{V}_{\boldsymbol{\omega}^*}(s) - \widetilde{V}^{\pi_{\boldsymbol{\theta}}}(s) \right)^2 \right]} \leq \sqrt{\gamma} \epsilon \leq \epsilon.
$$

Therefore, $LHS \leq 2\sqrt{2}\epsilon$.

**Proof of Proposition 4.5**  For any $\boldsymbol{\theta}_1, \boldsymbol{\theta}_2 \in \Omega(\boldsymbol{\theta})$, let

$$f(a) = \begin{cases} 1 & , \pi_{\boldsymbol{\theta}_1}(a|s) \geq \pi_{\boldsymbol{\theta}_2}(a|s) \\ -1 & , \text{otherwise} \end{cases},$$

then

$$\|\pi_{\boldsymbol{\theta}_1}(\cdot|s) - \pi_{\boldsymbol{\theta}_2}(\cdot|s)\|_1 = \mathbb{E}_{a \sim \pi_{\boldsymbol{\theta}_1}(\cdot|s)}[f(a)] - \mathbb{E}_{a \sim \pi_{\boldsymbol{\theta}_2}(\cdot|s)}[f(a)].$$

Note that

$$\begin{aligned}
\left\|\nabla_{\boldsymbol{\theta}} \mathbb{E}_{a \sim \pi_{\boldsymbol{\theta}}(\cdot|s)}[f(a)]\right\|_2 &= \left\|\nabla_{\boldsymbol{\theta}} \int_{\mathcal{A}} \pi_{\boldsymbol{\theta}}(a|s)f(a)\mathrm{d}a\right\|_2 \\
&= \left\|\int_{\mathcal{A}} \nabla_{\boldsymbol{\theta}} \pi_{\boldsymbol{\theta}}(a|s)f(a)\mathrm{d}a\right\|_2 \\
&= \left\|\int_{\mathcal{A}} \pi_{\boldsymbol{\theta}}(a|s) \nabla_{\boldsymbol{\theta}} \log \pi_{\boldsymbol{\theta}}(a|s)f(a)\mathrm{d}a\right\|_2 \\
&= \left\|\mathbb{E}_{a \sim \pi_{\boldsymbol{\theta}}(\cdot|s)}[\nabla_{\boldsymbol{\theta}} \log \pi_{\boldsymbol{\theta}}(a|s)f(a)]\right\|_2 \\
&\leq \mathbb{E}_{a \sim \pi_{\boldsymbol{\theta}}(\cdot|s)}[\|\nabla_{\boldsymbol{\theta}} \log \pi_{\boldsymbol{\theta}}(a|s)\|_2 |f(a)|] \\
&\leq L.
\end{aligned}$$

Therefore,

$$\|\pi_{\boldsymbol{\theta}_1}(\cdot|s) - \pi_{\boldsymbol{\theta}_2}(\cdot|s)\|_1 \leq L\|\boldsymbol{\theta}_1 - \boldsymbol{\theta}_2\|.$$

**Proof of Corollary 4.8**  Assume that $\mathbb{E}\|h_{\boldsymbol{\varphi}}(t)\|_2^2 \leq C_{\boldsymbol{\varphi}}^2$ and $\eta_t^{\varphi} = \frac{c_{\boldsymbol{\varphi}}}{\sqrt{t}}$, we have

$$\begin{aligned}
F_T &= \frac{1}{T/2} \sum_{t=T/2}^{T-1} \mathbb{E}\|\boldsymbol{\varphi}_{t+1} - \boldsymbol{\varphi}_t\|_2^2 \\
&= \frac{1}{T/2} \sum_{t=T/2}^{T-1} \eta_t^{\varphi 2} \mathbb{E}\|h_{\boldsymbol{\varphi}}(t)\|_2^2 \\
&\leq \frac{C_{\boldsymbol{\varphi}}^2}{T/2} \sum_{t=T/2}^{T-1} \frac{c_{\boldsymbol{\varphi}}^2}{t} \\
&= O(1/T)
\end{aligned}$$

Hence, the terms $O(F_T\sqrt{T})$ and $O(\sqrt{F_T/T})$ are both dominated by $O(1/\sqrt{T})$, leading to an overall $O(1/\sqrt{T}) + O(\epsilon)$ bound.

**Proof of Proposition 4.9**  We abuse the notation $\widehat{\mathcal{P}}_{\boldsymbol{\theta}} : \Delta(\mathcal{S}) \to \Delta(\mathcal{S})$ to denote an operator that acts on a state distribution $\nu$, defined by

$$\begin{aligned}
(\widehat{\mathcal{P}}_{\boldsymbol{\theta}}\nu)(s') &= \int_{\mathcal{S}} \mathrm{d}s \int_{\mathcal{A}} \mathrm{d}a \nu(s) \pi_{\boldsymbol{\theta}}(a|s) \widehat{\mathcal{P}}(s'|s,a) \\
&= \gamma \int_{\mathcal{S}} \mathrm{d}s \int_{\mathcal{A}} \mathrm{d}a \nu(s) \pi_{\boldsymbol{\theta}}(a|s) \mathcal{P}(s'|s,a) + (1-\gamma)\rho(s')
\end{aligned}$$

Then, $\widehat{\mathcal{P}}_{\boldsymbol{\theta}}$ is a contraction mapping and $\nu_{\rho}^{\pi_{\boldsymbol{\theta}}}$ is the unique fix point of it. Formally, $\forall \nu_1, \nu_2 \in \Delta(\mathcal{S})$, we have

$$\begin{aligned}
\left\|\widehat{\mathcal{P}}_{\boldsymbol{\theta}}\nu_1 - \widehat{\mathcal{P}}_{\boldsymbol{\theta}}\nu_2\right\|_1 &= \int_{\mathcal{S}} \mathrm{d}s' \left|(\widehat{\mathcal{P}}_{\boldsymbol{\theta}}\nu_1)(s') - (\widehat{\mathcal{P}}_{\boldsymbol{\theta}}\nu_2)(s')\right| \\
&= \gamma \int_{\mathcal{S}} \mathrm{d}s' \left|\int_{\mathcal{S}} \mathrm{d}s \int_{\mathcal{A}} \mathrm{d}a (\nu_1(s) - \nu_2(s)) \pi_{\boldsymbol{\theta}}(a|s) \mathcal{P}(s'|s,a)\right| \\
&\leq \gamma \int_{\mathcal{S}} \mathrm{d}s |\nu_1(s) - \nu_2(s)| \int_{\mathcal{S}} \mathrm{d}s' \int_{\mathcal{A}} \mathrm{d}a \pi_{\boldsymbol{\theta}}(a|s) \mathcal{P}(s'|s,a)
\end{aligned}$$

$$=\gamma\|\nu_1-\nu_2\|_1,$$

and

$$(\widehat{\mathcal{P}}_{\boldsymbol{\theta}}\nu_\rho^{\pi_{\boldsymbol{\theta}}})(s)=\nu_\rho^{\pi_{\boldsymbol{\theta}}}(s),\forall s\in\mathcal{S}.$$

Therefore,

$$
\begin{aligned}
\mathbb{E}\|\hat{\nu}_t-\nu_\rho^{\pi_{\boldsymbol{\theta}_t}}\|_1 &\leq \mathbb{E}\|\hat{\nu}_t-\nu_\rho^{\pi_{\boldsymbol{\theta}_{t-1}}}\|_1+\mathbb{E}\|\nu_\rho^{\pi_{\boldsymbol{\theta}_{t-1}}}-\nu_\rho^{\pi_{\boldsymbol{\theta}_t}}\|_1 \\
&\leq \mathbb{E}\left\|\widehat{\mathcal{P}}_{\boldsymbol{\theta}_{t-1}}\hat{\nu}_{t-1}-\widehat{\mathcal{P}}_{\boldsymbol{\theta}_{t-1}}\nu_\rho^{\pi_{\boldsymbol{\theta}_{t-1}}}\right\|_1+L_\nu\mathbb{E}\|\boldsymbol{\theta}_{t-1}-\boldsymbol{\theta}_t\|_2 \\
&\leq \gamma\mathbb{E}\|\hat{\nu}_{t-1}-\nu_\rho^{\pi_{\boldsymbol{\theta}_{t-1}}}\|_1+LC_\delta L_\nu\eta_{t-1}^{\boldsymbol{\theta}} \\
&\leq LC_\delta L_\nu\sum_{k=0}^{t-1}\gamma^{t-1-k}\eta_k^{\boldsymbol{\theta}}+\gamma^t\|\rho-\nu_\rho^{\pi_{\boldsymbol{\theta}_0}}\|_1.
\end{aligned}
$$

**Proof of Lemma B.1**

$$
\begin{aligned}
\left|\widetilde{V}_{\boldsymbol{\varphi}}^{\pi_{\boldsymbol{\theta}}}(s)\right| &=\left|\frac{1}{1-\gamma}\mathbb{E}_{s\sim\nu_\rho^{\pi_{\boldsymbol{\theta}}}(\cdot)}\left[\mathbb{E}_{a\sim\pi_{\boldsymbol{\theta}}(\cdot|s)}\left[\tilde{r}_{\boldsymbol{\varphi},\boldsymbol{\theta}}(s,a)\right]\right]\right| \\
&\leq \frac{1}{1-\gamma}\mathbb{E}_{s\sim\nu_\rho^{\pi_{\boldsymbol{\theta}}}(\cdot)}\left[\left|\mathbb{E}_{a\sim\pi_{\boldsymbol{\theta}}(\cdot|s)}\left[\tilde{r}_{\boldsymbol{\varphi},\boldsymbol{\theta}}(s,a)\right]\right|\right] \\
&\leq \frac{C}{1-\gamma}
\end{aligned}
$$

Hence, $C_J=O((1-\gamma)^{-1})$. Note that by letting $\rho(s)=\mathbb{I}[s=s_0]$, we have $\left|\widetilde{V}_{\boldsymbol{\varphi}}^{\pi_{\boldsymbol{\theta}}}(s_0)\right|\leq C_J$ for any $s_0\in\mathcal{S}$.

$$
\begin{aligned}
\|\nabla_{\boldsymbol{\theta}}J_{\boldsymbol{\varphi}}(\boldsymbol{\theta})\|_2 &=\left\|\frac{1}{1-\gamma}\mathbb{E}_{s\sim\nu_\rho^{\pi_{\boldsymbol{\theta}}}(\cdot)}\left[\mathbb{E}_{a\sim\pi_{\boldsymbol{\theta}}(\cdot|s)}\left[\widetilde{Q}_{\boldsymbol{\varphi}}^{\pi_{\boldsymbol{\theta}}}(s,a)\nabla_{\boldsymbol{\theta}}\log\pi_{\boldsymbol{\theta}}(a|s)\right]\right]\right\|_2 \\
&\leq \frac{L}{1-\gamma}\mathbb{E}_{s\sim\nu_\rho^{\pi_{\boldsymbol{\theta}}}(\cdot),a\sim\pi_{\boldsymbol{\theta}}(\cdot|s)}\left|\tilde{r}(s,a)+\gamma\mathbb{E}_{s'\sim\mathcal{P}(\cdot|s,a)}\left[\widetilde{V}_{\boldsymbol{\varphi}}^{\pi_{\boldsymbol{\theta}}}(s)\right]\right| \\
&\leq \frac{CL}{(1-\gamma)^2}
\end{aligned}
$$

Hence, $L_J=O((1-\gamma)^{-2})$. Similarly, by letting $\rho(s)=\mathbb{I}[s=s_0]$, we have $\left|\nabla_{\boldsymbol{\theta}}\widetilde{V}_{\boldsymbol{\varphi}}^{\pi_{\boldsymbol{\theta}}}(s_0)\right|\leq L_J$ for any $s_0\in\mathcal{S}$.

$$
\begin{aligned}
\nabla_{\boldsymbol{\theta}}^2J_{\boldsymbol{\varphi}}(\boldsymbol{\theta}) &=\frac{1}{1-\gamma}\mathbb{E}_{\nu_\rho^{\pi_{\boldsymbol{\theta}}},\pi_{\boldsymbol{\theta}}}\left[\widetilde{Q}^{\pi_{\boldsymbol{\theta}}}(s,a)\left(\nabla_{\boldsymbol{\theta}}\log\pi_{\boldsymbol{\theta}}(a|s)\nabla_{\boldsymbol{\theta}}\log\pi_{\boldsymbol{\theta}}(a|s)^\top+\nabla_{\boldsymbol{\theta}}^2\log\pi_{\boldsymbol{\theta}}(a|s)\right)\right] \\
&\quad+\frac{\gamma}{1-\gamma}\mathbb{E}_{\nu_\rho^{\pi_{\boldsymbol{\theta}}},\pi_{\boldsymbol{\theta}},\mathcal{P}}\left[\nabla_{\boldsymbol{\theta}}\log\pi_{\boldsymbol{\theta}}(a|s)\nabla_{\boldsymbol{\theta}}\widetilde{V}_{\boldsymbol{\varphi}}^{\pi_{\boldsymbol{\theta}}}(s')^\top+\nabla_{\boldsymbol{\theta}}\widetilde{V}_{\boldsymbol{\varphi}}^{\pi_{\boldsymbol{\theta}}}(s')\nabla_{\boldsymbol{\theta}}\log\pi_{\boldsymbol{\theta}}(a|s)^\top\right]
\end{aligned}
$$

$$\left\|\nabla_{\boldsymbol{\theta}}^2J_{\boldsymbol{\varphi}}(\boldsymbol{\theta})\right\|_2\leq\frac{C_J(L^2+S)}{1-\gamma}+\frac{2\gamma LL_J}{1-\gamma}=O((1-\gamma)^{-3}).$$

Hence, $S_J=O((1-\gamma)^{-3})$.

$$
\begin{aligned}
\|\nabla_{\boldsymbol{\varphi}}J_{\boldsymbol{\varphi}}(\boldsymbol{\theta})\|_2 &=\left\|\nabla_{\boldsymbol{\varphi}}\left(\frac{1}{1-\gamma}\mathbb{E}_{\nu_\rho^{\pi_{\boldsymbol{\theta}}},\pi_{\boldsymbol{\theta}}}\left[\tilde{r}_{\boldsymbol{\varphi},\boldsymbol{\theta}}(s,a)\right]\right)\right\|_2 \\
&=\left\|\frac{1}{1-\gamma}\mathbb{E}_{s\sim\nu_\rho^{\pi_{\boldsymbol{\theta}}}(\cdot)}\left[\nabla_{\boldsymbol{\varphi}}\mathbb{E}_{a\sim\pi_{\boldsymbol{\theta}}(\cdot|s)}\left[\tilde{r}_{\boldsymbol{\varphi},\boldsymbol{\theta}}(s,a)\right]\right]\right\|_2 \\
&\leq \frac{D}{1-\gamma}
\end{aligned}
$$

Hence, $D_J=O((1-\gamma)^{-1})$.

**Proof of Corollary B.2** For any state $\boldsymbol{\theta}_1, \boldsymbol{\theta}_2 \in \Omega(\boldsymbol{\theta})$, consider the MDP $(\mathcal{S}, \mathcal{A}, \mathcal{P}, r', \gamma)$ where for any $s \in \mathcal{S}, a \in \mathcal{A}$,

$$r'(s,a) = f(s) := \begin{cases} 1 & , \nu_\rho^{\pi_{\boldsymbol{\theta}_1}}(s) > \nu_\rho^{\pi_{\boldsymbol{\theta}_2}}(s) \\ -1 & , \text{otherwise} \end{cases}.$$

Assume the regularization factor $\alpha = 0$, then for this RL problem, $\tilde{r}(s,a) = r'(s,a)$, and

$$\begin{aligned}
\left\| \nu_\rho^{\pi_{\boldsymbol{\theta}_1}} - \nu_\rho^{\pi_{\boldsymbol{\theta}_2}} \right\|_1 &= \int_{\mathcal{S}} \mathrm{d}s \left| \nu_\rho^{\pi_{\boldsymbol{\theta}_1}}(s) - \nu_\rho^{\pi_{\boldsymbol{\theta}_2}}(s) \right| \\
&= \int_{\mathcal{S}} \mathrm{d}s \left( \nu_\rho^{\pi_{\boldsymbol{\theta}_1}}(s) - \nu_\rho^{\pi_{\boldsymbol{\theta}_2}}(s) \right) f(s) \\
&= \int_{\mathcal{S}} \mathrm{d}s \nu_\rho^{\pi_{\boldsymbol{\theta}_1}}(s) f(s) - \int_{\mathcal{S}} \mathrm{d}s \nu_\rho^{\pi_{\boldsymbol{\theta}_2}}(s) f(s) \\
&= \mathbb{E}_{s \sim \nu_\rho^{\pi_{\boldsymbol{\theta}_1}}(\cdot)} \left[ \mathbb{E}_{a \sim \pi_{\boldsymbol{\theta}_1}(\cdot|s)}[\tilde{r}(s,a)] \right] - \mathbb{E}_{s \sim \nu_\rho^{\pi_{\boldsymbol{\theta}_2}}(\cdot)} \left[ \mathbb{E}_{a \sim \pi_{\boldsymbol{\theta}_2}(\cdot|s)}[\tilde{r}(s,a)] \right] \\
&= (1-\gamma)(J(\boldsymbol{\theta}_1) - J(\boldsymbol{\theta}_2)).
\end{aligned}$$

Then we can apply Lemma B.1 with $C = 1$ to obtain $L_\nu = (1-\gamma)L_J$ and $S_\nu = (1-\gamma)S_J$.

**Proof of Lemma B.3**

$$\begin{aligned}
\|\nabla_{\boldsymbol{\theta}} \boldsymbol{A}_{\boldsymbol{\theta}}\|_2 &= \left\| \nabla_{\boldsymbol{\theta}} \mathbb{E}_{\nu_\rho^{\pi_{\boldsymbol{\theta}}}, \pi_{\boldsymbol{\theta}}, \mathcal{P}} \left[ \phi(s)(\phi(s) - \gamma\phi(s'))^\top \right] \right\|_2 \\
&= \left\| \int_{\mathcal{S}} \mathrm{d}s \int_{\mathcal{A}} \mathrm{d}a \nabla_{\boldsymbol{\theta}} \left( \nu_\rho^{\pi_{\boldsymbol{\theta}}}(s)\pi_{\boldsymbol{\theta}}(a|s) \right) \mathbb{E}_{s' \sim \mathcal{P}(\cdot|s,a)} \left[ \phi(s)(\phi(s) - \gamma\phi(s'))^\top \right] \right\|_2 \\
&\leq \left( \int_{\mathcal{S}} \mathrm{d}s \int_{\mathcal{A}} \mathrm{d}a \left\| \nabla_{\boldsymbol{\theta}} \left( \nu_\rho^{\pi_{\boldsymbol{\theta}}}(s)\pi_{\boldsymbol{\theta}}(a|s) \right) \right\|_2 \right) \left( \max_s \left\| \mathbb{E}_{s' \sim \mathcal{P}(\cdot|s,a)} \left[ \phi(s)(\phi(s) - \gamma\phi(s'))^\top \right] \right\|_2 \right) \\
&\leq (1+\gamma) \int_{\mathcal{S}} \mathrm{d}s \int_{\mathcal{A}} \mathrm{d}a \left\| \nabla_{\boldsymbol{\theta}} \left( \nu_\rho^{\pi_{\boldsymbol{\theta}}}(s)\pi_{\boldsymbol{\theta}}(a|s) \right) \right\|_2 \\
&= (1+\gamma) \int_{\mathcal{S}} \mathrm{d}s \int_{\mathcal{A}} \mathrm{d}a \left\| \nabla_{\boldsymbol{\theta}} \nu_\rho^{\pi_{\boldsymbol{\theta}}}(s)\pi_{\boldsymbol{\theta}}(a|s) + \nu_\rho^{\pi_{\boldsymbol{\theta}}}(s)\nabla_{\boldsymbol{\theta}}\pi_{\boldsymbol{\theta}}(a|s) \right\|_2 \\
&\leq (1+\gamma) \left( \int_{\mathcal{S}} \mathrm{d}s \left\| \nabla_{\boldsymbol{\theta}} \nu_\rho^{\pi_{\boldsymbol{\theta}}}(s) \right\|_2 \int_{\mathcal{A}} \mathrm{d}a \pi_{\boldsymbol{\theta}}(a|s) + \int_{\mathcal{S}} \mathrm{d}s \nu_\rho^{\pi_{\boldsymbol{\theta}}}(s) \int_{\mathcal{A}} \mathrm{d}a \pi_{\boldsymbol{\theta}}(a|s) \left\| \nabla_{\boldsymbol{\theta}} \log \pi_{\boldsymbol{\theta}}(a|s) \right\|_2 \right) \\
&\leq (1+\gamma)(L_\mu + L) \\
&= O((1-\gamma)^{-1})
\end{aligned}$$

Hence, $L_A = O((1-\gamma)^{-1})$.

$$\begin{aligned}
\left\| \nabla_{\boldsymbol{\theta}}^2 \boldsymbol{A}_{\boldsymbol{\theta}} \right\|_2 &= \left\| \nabla_{\boldsymbol{\theta}}^2 \mathbb{E}_{\nu_\rho^{\pi_{\boldsymbol{\theta}}}, \pi_{\boldsymbol{\theta}}, \mathcal{P}} \left[ \phi(s)(\phi(s) - \gamma\phi(s'))^\top \right] \right\|_2 \\
&= \left\| \int_{\mathcal{S}} \mathrm{d}s \int_{\mathcal{A}} \mathrm{d}a \nabla_{\boldsymbol{\theta}}^2 \left( \nu_\rho^{\pi_{\boldsymbol{\theta}}}(s)\pi_{\boldsymbol{\theta}}(a|s) \right) \mathbb{E}_{s' \sim \mathcal{P}(\cdot|s,a)} \left[ \phi(s)(\phi(s) - \gamma\phi(s'))^\top \right] \right\|_2 \\
&\leq \left( \int_{\mathcal{S}} \mathrm{d}s \int_{\mathcal{A}} \mathrm{d}a \left\| \nabla_{\boldsymbol{\theta}}^2 \left( \nu_\rho^{\pi_{\boldsymbol{\theta}}}(s)\pi_{\boldsymbol{\theta}}(a|s) \right) \right\|_2 \right) \left( \max_s \left\| \mathbb{E}_{s' \sim \mathcal{P}(\cdot|s,a)} \left[ \phi(s)(\phi(s) - \gamma\phi(s'))^\top \right] \right\|_2 \right) \\
&\leq (1+\gamma) \int_{\mathcal{S}} \mathrm{d}s \int_{\mathcal{A}} \mathrm{d}a \left\| \nabla_{\boldsymbol{\theta}}^2 \left( \nu_\rho^{\pi_{\boldsymbol{\theta}}}(s)\pi_{\boldsymbol{\theta}}(a|s) \right) \right\|_2 \\
&\leq (1+\gamma) \left[ \int_{\mathcal{S}} \mathrm{d}s \left\| \nabla_{\boldsymbol{\theta}}^2 \nu_\rho^{\pi_{\boldsymbol{\theta}}}(s) \right\|_2 \int_{\mathcal{A}} \mathrm{d}a \pi_{\boldsymbol{\theta}}(a|s) \right. \\
&\quad + 2 \int_{\mathcal{S}} \mathrm{d}s \left\| \nabla_{\boldsymbol{\theta}} \nu_\rho^{\pi_{\boldsymbol{\theta}}}(s) \right\|_2 \int_{\mathcal{A}} \mathrm{d}a \pi_{\boldsymbol{\theta}}(a|s) \left\| \nabla_{\boldsymbol{\theta}} \log \pi_{\boldsymbol{\theta}}(a|s) \right\|_2 \\
&\quad \left. + \int_{\mathcal{S}} \mathrm{d}s \nu_\rho^{\pi_{\boldsymbol{\theta}}}(s) \int_{\mathcal{A}} \mathrm{d}a \pi_{\boldsymbol{\theta}}(a|s) \left\| \nabla_{\boldsymbol{\theta}} \log \pi_{\boldsymbol{\theta}}(a|s) \nabla_{\boldsymbol{\theta}} \log \pi_{\boldsymbol{\theta}}(a|s)^\top + \nabla_{\boldsymbol{\theta}}^2 \log \pi_{\boldsymbol{\theta}}(a|s) \right\|_2 \right] \\
&\leq (1+\gamma) \left( S_\nu + 2LL_\nu + L^2 + S \right) \\
&= O((1-\gamma)^{-2})
\end{aligned}$$

Hence, $S_A = O((1-\gamma)^{-2})$.

**Proof of Lemma B.4**

$$\|\boldsymbol{b}_{\boldsymbol{\varphi},\boldsymbol{\theta}}\|_2 = \left\|\mathbb{E}_{\nu_\rho^{\pi_\theta},\pi_\theta}[\tilde{r}_{\boldsymbol{\varphi},\boldsymbol{\theta}}(s,a)\phi(s)]\right\|_2$$

$$\leq \left|\mathbb{E}_{\nu_\rho^{\pi_\theta},\pi_\theta}[\tilde{r}_{\boldsymbol{\varphi},\boldsymbol{\theta}}(s,a)]\right|$$

$$\leq C$$

Hence, $C_b = O(1)$.

$$\|\nabla_{\boldsymbol{\theta}}\boldsymbol{b}_{\boldsymbol{\varphi},\boldsymbol{\theta}}\|_2 = \left\|\nabla_{\boldsymbol{\theta}}\mathbb{E}_{\nu_\rho^{\pi_\theta},\pi_\theta}[\tilde{r}_{\boldsymbol{\varphi},\boldsymbol{\theta}}(s,a)\phi(s)]\right\|_2$$

$$= \left\|\int_{\mathcal{S}}\mathrm{d}s\nabla_{\boldsymbol{\theta}}\left(\nu_\rho^{\pi_\theta}(s)\mathbb{E}_{a\sim\pi_\theta(\cdot|s)}[\tilde{r}_{\boldsymbol{\varphi},\boldsymbol{\theta}}(s,a)]\right)\phi(s)\right\|_2$$

$$\leq \left\|\int_{\mathcal{S}}\mathrm{d}s\nabla_{\boldsymbol{\theta}}\left(\nu_\rho^{\pi_\theta}(s)\mathbb{E}_{a\sim\pi_\theta(\cdot|s)}[\tilde{r}_{\boldsymbol{\varphi},\boldsymbol{\theta}}(s,a)]\right)\right\|_2\left(\max_s\|\phi(s)\|_2\right)$$

$$\leq \int_{\mathcal{S}}\mathrm{d}s\left\|\nabla_{\boldsymbol{\theta}}\left(\nu_\rho^{\pi_\theta}(s)\mathbb{E}_{a\sim\pi_\theta(\cdot|s)}[\tilde{r}_{\boldsymbol{\varphi},\boldsymbol{\theta}}(s,a)]\right)\right\|_2$$

$$\leq \int_{\mathcal{S}}\mathrm{d}s\left\|\nabla_{\boldsymbol{\theta}}\nu_\rho^{\pi_\theta}(s)\right\|_2\mathbb{E}_{a\sim\pi_\theta(\cdot|s)}[\tilde{r}_{\boldsymbol{\varphi},\boldsymbol{\theta}}(s,a)]$$

$$+ \int_{\mathcal{S}}\mathrm{d}s\nu_\rho^{\pi_\theta}(s)\left\|\nabla_{\boldsymbol{\theta}}\mathbb{E}_{a\sim\pi_\theta(\cdot|s)}[\tilde{r}_{\boldsymbol{\varphi},\boldsymbol{\theta}}(s,a)]\right\|_2$$

$$\leq CL_\nu + CL + 0$$

$$= O((1-\gamma)^{-1})$$

Hence, $L_b = O((1-\gamma)^{-1})$.

$$\left\|\nabla_{\boldsymbol{\theta}}^2\boldsymbol{b}_{\boldsymbol{\varphi},\boldsymbol{\theta}}\right\|_2 = \left\|\nabla_{\boldsymbol{\theta}}^2\mathbb{E}_{\nu_\rho^{\pi_\theta},\pi_\theta}[\tilde{r}_{\boldsymbol{\varphi},\boldsymbol{\theta}}(s,a)\phi(s)]\right\|_2$$

$$= \left\|\int_{\mathcal{S}}\mathrm{d}s\nabla_{\boldsymbol{\theta}}^2\left(\nu_\rho^{\pi_\theta}(s)\mathbb{E}_{a\sim\pi_\theta(\cdot|s)}[\tilde{r}_{\boldsymbol{\varphi},\boldsymbol{\theta}}(s,a)]\right)\phi(s)\right\|_2$$

$$\leq \left\|\int_{\mathcal{S}}\mathrm{d}s\nabla_{\boldsymbol{\theta}}^2\left(\nu_\rho^{\pi_\theta}(s)\mathbb{E}_{a\sim\pi_\theta(\cdot|s)}[\tilde{r}_{\boldsymbol{\varphi},\boldsymbol{\theta}}(s,a)]\right)\right\|_2\left(\max_s\|\phi(s)\|_2\right)$$

$$\leq \int_{\mathcal{S}}\mathrm{d}s\left\|\nabla_{\boldsymbol{\theta}}^2\left(\nu_\rho^{\pi_\theta}(s)\mathbb{E}_{a\sim\pi_\theta(\cdot|s)}[\tilde{r}_{\boldsymbol{\varphi},\boldsymbol{\theta}}(s,a)]\right)\right\|_2$$

$$\leq \int_{\mathcal{S}}\mathrm{d}s\left\|\nabla_{\boldsymbol{\theta}}^2\nu_\rho^{\pi_\theta}(s)\right\|_2\mathbb{E}_{a\sim\pi_\theta(\cdot|s)}[\tilde{r}_{\boldsymbol{\varphi},\boldsymbol{\theta}}(s,a)]$$

$$+ \int_{\mathcal{S}}\mathrm{d}s\left\|\nabla_{\boldsymbol{\theta}}\nu_\rho^{\pi_\theta}(s)\right\|_2\left\|\nabla_{\boldsymbol{\theta}}\mathbb{E}_{a\sim\pi_\theta(\cdot|s)}[\tilde{r}_{\boldsymbol{\varphi},\boldsymbol{\theta}}(s,a)]\right\|_2$$

$$+ \int_{\mathcal{S}}\mathrm{d}s\nu_\rho^{\pi_\theta}(s)\left\|\nabla_{\boldsymbol{\theta}}^2\mathbb{E}_{a\sim\pi_\theta(\cdot|s)}[\tilde{r}_{\boldsymbol{\varphi},\boldsymbol{\theta}}(s,a)]\right\|_2$$

$$\leq CS_\nu + CLL_\nu + C(L^2 + S)$$

$$= O((1-\gamma)^{-2})$$

Hence, $S_b = O((1-\gamma)^{-2})$.

$$\|\nabla_{\boldsymbol{\varphi}}\boldsymbol{b}_{\boldsymbol{\varphi},\boldsymbol{\theta}}\|_2 = \left\|\nabla_{\boldsymbol{\varphi}}\mathbb{E}_{\nu_\rho^{\pi_{\boldsymbol{\theta}}},\pi_{\boldsymbol{\theta}}}[\tilde{r}_{\boldsymbol{\varphi},\boldsymbol{\theta}}(s,a)\phi(s)]\right\|_2$$

$$= \left\|\int_{\mathcal{S}}\mathrm{d}s\nabla_{\boldsymbol{\varphi}}\left(\nu_\rho^{\pi_{\boldsymbol{\theta}}}(s)\mathbb{E}_{a\sim\pi_{\boldsymbol{\theta}}(\cdot|s)}\left[\tilde{r}_{\boldsymbol{\varphi},\boldsymbol{\theta}}(s,a)\right]\right)\phi(s)\right\|_2$$

$$\leq \left\|\int_{\mathcal{S}}\mathrm{d}s\nabla_{\boldsymbol{\varphi}}\left(\nu_\rho^{\pi_{\boldsymbol{\theta}}}(s)\mathbb{E}_{a\sim\pi_{\boldsymbol{\theta}}(\cdot|s)}\left[\tilde{r}_{\boldsymbol{\varphi},\boldsymbol{\theta}}(s,a)\right]\right)\right\|_2\left(\max_s\|\phi(s)\|_2\right)$$

$$\leq \int_{\mathcal{S}}\mathrm{d}s\left\|\nabla_{\boldsymbol{\varphi}}\left(\nu_\rho^{\pi_{\boldsymbol{\theta}}}(s)\mathbb{E}_{a\sim\pi_{\boldsymbol{\theta}}(\cdot|s)}\left[\tilde{r}_{\boldsymbol{\varphi},\boldsymbol{\theta}}(s,a)\right]\right)\right\|_2$$

$$= \int_{\mathcal{S}}\mathrm{d}s\nu_\rho^{\pi_{\boldsymbol{\theta}}}(s)\left\|\nabla_{\boldsymbol{\varphi}}\mathbb{E}_{a\sim\pi_{\boldsymbol{\theta}}(\cdot|s)}\left[\tilde{r}_{\boldsymbol{\varphi},\boldsymbol{\theta}}(s,a)\right]\right\|_2$$

$$\leq D$$

Hence, $D_b = O(1)$.

**Proof of Lemma B.5**

$$\|\boldsymbol{\omega}^*(\boldsymbol{\varphi},\boldsymbol{\theta})\|_2 = \|\boldsymbol{A}_{\boldsymbol{\theta}}^{-1}\boldsymbol{b}_{\boldsymbol{\varphi},\boldsymbol{\theta}}\|_2 \leq \|\boldsymbol{A}_{\boldsymbol{\theta}}^{-1}\|_2\|\boldsymbol{b}_{\boldsymbol{\varphi},\boldsymbol{\theta}}\|_2 \leq \frac{C_b}{\lambda} = \frac{C}{\lambda}$$

Hence, $C_{\boldsymbol{\omega}} = O(\lambda^{-1})$.

$$\|\nabla_{\boldsymbol{\theta}}\boldsymbol{\omega}^*(\boldsymbol{\varphi},\boldsymbol{\theta})\|_2 = \left\|\nabla_{\boldsymbol{\theta}}\left(\boldsymbol{A}_{\boldsymbol{\theta}}^{-1}\boldsymbol{b}_{\boldsymbol{\varphi},\boldsymbol{\theta}}\right)\right\|_2$$

$$= \left\|\nabla_{\boldsymbol{\theta}}\left(\boldsymbol{A}_{\boldsymbol{\theta}}^{-1}\right)\boldsymbol{b}_{\boldsymbol{\varphi},\boldsymbol{\theta}} + \boldsymbol{A}_{\boldsymbol{\theta}}^{-1}\nabla_{\boldsymbol{\theta}}\boldsymbol{b}_{\boldsymbol{\varphi},\boldsymbol{\theta}}\right\|_2$$

$$= \left\|\boldsymbol{A}_{\boldsymbol{\theta}}^{-1}\nabla_{\boldsymbol{\theta}}\boldsymbol{A}_{\boldsymbol{\theta}}\boldsymbol{A}_{\boldsymbol{\theta}}^{-1}\boldsymbol{b}_{\boldsymbol{\varphi},\boldsymbol{\theta}} + \boldsymbol{A}_{\boldsymbol{\theta}}^{-1}\nabla_{\boldsymbol{\theta}}\boldsymbol{b}_{\boldsymbol{\varphi},\boldsymbol{\theta}}\right\|_2$$

$$\leq \|\boldsymbol{A}_{\boldsymbol{\theta}}^{-1}\|_2\|\nabla_{\boldsymbol{\theta}}\boldsymbol{A}_{\boldsymbol{\theta}}\|_2\|\boldsymbol{A}_{\boldsymbol{\theta}}^{-1}\|_2\|\boldsymbol{b}_{\boldsymbol{\varphi},\boldsymbol{\theta}}\|_2 + \|\boldsymbol{A}_{\boldsymbol{\theta}}^{-1}\|_2\|\nabla_{\boldsymbol{\theta}}\boldsymbol{b}_{\boldsymbol{\varphi},\boldsymbol{\theta}}\|_2$$

$$\leq L_A C_b \lambda^{-2} + L_b \lambda^{-1}$$

$$= O((1-\gamma)^{-1}\lambda^{-2})$$

Hence, $L_{\boldsymbol{\omega}} = O((1-\gamma)^{-1}\lambda^{-2})$.

$$\left\|\nabla_{\boldsymbol{\theta}}^2\boldsymbol{\omega}^*(\boldsymbol{\varphi},\boldsymbol{\theta})\right\|_2 = \left\|\nabla_{\boldsymbol{\theta}}^2\left(\boldsymbol{A}_{\boldsymbol{\theta}}^{-1}\boldsymbol{b}_{\boldsymbol{\varphi},\boldsymbol{\theta}}\right)\right\|_2$$

$$= \left\|\nabla_{\boldsymbol{\theta}}\left(\boldsymbol{A}_{\boldsymbol{\theta}}^{-1}\nabla_{\boldsymbol{\theta}}\boldsymbol{A}_{\boldsymbol{\theta}}\boldsymbol{A}_{\boldsymbol{\theta}}^{-1}\boldsymbol{b}_{\boldsymbol{\varphi},\boldsymbol{\theta}} + \boldsymbol{A}_{\boldsymbol{\theta}}^{-1}\nabla_{\boldsymbol{\theta}}\boldsymbol{b}_{\boldsymbol{\varphi},\boldsymbol{\theta}}\right)\right\|_2$$

$$= \left\|\boldsymbol{A}_{\boldsymbol{\theta}}^{-1}\nabla_{\boldsymbol{\theta}}^2\boldsymbol{A}_{\boldsymbol{\theta}}\boldsymbol{A}_{\boldsymbol{\theta}}^{-1}\boldsymbol{b}_{\boldsymbol{\varphi},\boldsymbol{\theta}} + 2\boldsymbol{A}_{\boldsymbol{\theta}}^{-1}\nabla_{\boldsymbol{\theta}}\boldsymbol{A}_{\boldsymbol{\theta}}\boldsymbol{A}_{\boldsymbol{\theta}}^{-1}\nabla_{\boldsymbol{\theta}}\boldsymbol{A}_{\boldsymbol{\theta}}\boldsymbol{A}_{\boldsymbol{\theta}}^{-1}\boldsymbol{b}_{\boldsymbol{\varphi},\boldsymbol{\theta}}\right.$$

$$\left. + 2\boldsymbol{A}_{\boldsymbol{\theta}}^{-1}\nabla_{\boldsymbol{\theta}}\boldsymbol{A}_{\boldsymbol{\theta}}\boldsymbol{A}_{\boldsymbol{\theta}}^{-1}\nabla_{\boldsymbol{\theta}}\boldsymbol{b}_{\boldsymbol{\varphi},\boldsymbol{\theta}} + \boldsymbol{A}_{\boldsymbol{\theta}}^{-1}\nabla_{\boldsymbol{\theta}}^2\boldsymbol{b}_{\boldsymbol{\varphi},\boldsymbol{\theta}}\right\|_2$$

$$\leq \|\boldsymbol{A}_{\boldsymbol{\theta}}^{-1}\|_2\|\nabla_{\boldsymbol{\theta}}^2\boldsymbol{A}_{\boldsymbol{\theta}}\|_2\|\boldsymbol{A}_{\boldsymbol{\theta}}^{-1}\|_2\|\boldsymbol{b}_{\boldsymbol{\varphi},\boldsymbol{\theta}}\|_2$$

$$+ 2\|\boldsymbol{A}_{\boldsymbol{\theta}}^{-1}\|_2\|\nabla_{\boldsymbol{\theta}}\boldsymbol{A}_{\boldsymbol{\theta}}\|_2\|\boldsymbol{A}_{\boldsymbol{\theta}}^{-1}\|_2\|\nabla_{\boldsymbol{\theta}}\boldsymbol{A}_{\boldsymbol{\theta}}\|_2\|\boldsymbol{A}_{\boldsymbol{\theta}}^{-1}\|_2\|\boldsymbol{b}_{\boldsymbol{\varphi},\boldsymbol{\theta}}\|_2.$$

$$+ 2\|\boldsymbol{A}_{\boldsymbol{\theta}}^{-1}\|_2\|\nabla_{\boldsymbol{\theta}}\boldsymbol{A}_{\boldsymbol{\theta}}\|_2\|\boldsymbol{A}_{\boldsymbol{\theta}}^{-1}\|_2\|\nabla_{\boldsymbol{\theta}}\boldsymbol{b}_{\boldsymbol{\varphi},\boldsymbol{\theta}}\|_2.$$

$$+ \|\boldsymbol{A}_{\boldsymbol{\theta}}^{-1}\|_2\|\nabla_{\boldsymbol{\theta}}^2\boldsymbol{b}_{\boldsymbol{\varphi},\boldsymbol{\theta}}\|_2$$

$$\leq S_A C_b \lambda^{-2} + 2L_A^2 C_b \lambda^{-3} + 2L_A L_b \lambda^{-2} + S_b \lambda^{-1}$$

$$= O((1-\gamma)^{-2}\lambda^{-3})$$

Hence, $S_{\boldsymbol{\omega}} = O((1-\gamma)^{-2}\lambda^{-3})$.

$$\|\nabla_{\boldsymbol{\varphi}}\boldsymbol{\omega}^*(\boldsymbol{\varphi},\boldsymbol{\theta})\|_2 = \left\|\nabla_{\boldsymbol{\varphi}}\left(\boldsymbol{A}_{\boldsymbol{\theta}}^{-1}\boldsymbol{b}_{\boldsymbol{\varphi},\boldsymbol{\theta}}\right)\right\|_2$$

$$= \left\|\boldsymbol{A}_{\boldsymbol{\theta}}^{-1}\nabla_{\boldsymbol{\varphi}}\boldsymbol{b}_{\boldsymbol{\varphi},\boldsymbol{\theta}}\right\|_2$$

$$\leq \|\boldsymbol{A}_{\boldsymbol{\theta}}^{-1}\|_2\|\nabla_{\boldsymbol{\varphi}}\boldsymbol{b}_{\boldsymbol{\varphi},\boldsymbol{\theta}}\|_2$$

$$\leq \frac{D_b}{\lambda}$$

$$= O(\lambda^{-1})$$

Hence, $D_{\boldsymbol{\omega}} = O(\lambda^{-1})$.

**Proof of Lemma B.6**

$$\mathbb{E}_{\nu,\pi_{\boldsymbol{\theta}},\mathcal{P}}\|\hat{\delta}(s,a,s')\|_2^2 = \int_{\mathcal{S}} \mathrm{d}s\nu(s)\mathbb{E}_{\pi_{\boldsymbol{\theta}},\mathcal{P}}\left[\left(\tilde{r}(s,a) + (\phi(s') - \phi(s))^{\top}\boldsymbol{\omega}\right)^2\right]$$

$$\leq \int_{\mathcal{S}} \mathrm{d}s\nu(s)\mathbb{E}_{\pi_{\boldsymbol{\theta}}}\left[\left(\tilde{r}(s,a) + 2C_{\boldsymbol{\omega}}\right)^2\right]$$

$$\leq \int_{\mathcal{S}} \mathrm{d}s\nu(s)\left(\mathbb{E}_{\pi_{\boldsymbol{\theta}}}\left[\tilde{r}(s,a)^2\right] + 4C_{\boldsymbol{\omega}}\mathbb{E}_{\pi_{\boldsymbol{\theta}}}\left[\tilde{r}(s,a)\right] + 4C_{\boldsymbol{\omega}}^2\right)$$

$$\leq (C^2 + 4CC_{\boldsymbol{\omega}} + 4C_{\boldsymbol{\omega}}^2)$$

$$= (C + 2C_{\boldsymbol{\omega}})^2$$

Hence, $C_{\delta} = C + 2C_{\boldsymbol{\omega}} = O(\lambda^{-1})$.

# E  LLM USAGE

Large Language Models (LLMs) were used to aid in the writing and polishing of the manuscript. Specifically, we used an LLM to assist in refining the language, improving readability, and ensuring clarity in various sections of the paper. The model helped with tasks such as sentence rephrasing, grammar checking, and enhancing the overall flow of the text.

It is important to note that the LLM was not involved in the ideation, research methodology, or experimental design. All research concepts, ideas, and analyses were developed and conducted by the authors. The contributions of the LLM were solely focused on improving the linguistic quality of the paper, with no involvement in the scientific content or data analysis.

The authors take full responsibility for the content of the manuscript, including any text generated or polished by the LLM. We have ensured that the LLM-generated text adheres to ethical guidelines and does not contribute to plagiarism or scientific misconduct.

