# OpenReview forum: "Finite-time Convergence Analysis of Actor-Critic with Evolving Reward"
_ICLR.cc/2026/Conference — Submitted to ICLR 2026_

### Official Review · Reviewer_ziai · 2025-10-27

**Soundness:** 4
**Presentation:** 4
**Contribution:** 2
**Rating:** 4
**Confidence:** 4

**Summary:**

The paper provides a finite-time convergence analysis for single-timescale actor–critic algorithms when the reward function evolves over time. It proves that if the reward changes slowly (in a Lipschitz and bounded manner), the standard $O(1/\sqrt{T})$ convergence rate can still be achieved. The analysis extends known results for static rewards and introduces an improved treatment of distribution mismatch under Markovian sampling.

**Strengths:**

- The theoretical framework is well-specified and the proofs are systematic; paper is well presented and self-contained
- This work tackles an underexplored theoretical setting and the extension  of the single timescale actor-critic algorithm to evolving rewards bridges a theoretical gap
- Improved rate under Markovian sampling where the tighter control over distribution mismatch (shaving off $\log ^2 T$) is a nice refinement, even if tangential to the main theme

**Weaknesses:**

- The paper does a poor job of positioning the significance of its work in comparison with prior literature on theoretical analysis with changing dynamics in non-stationary RL, adversarial RL and performative RL (see Questions below)
- The evolving reward model considered here is too restrictive. The paper considers rewards evolve smoothly and slowly, whereas real-world scenarios involve abrupt or feedback-driven changes
- I believe the proof techniques have limited theoretical challenge (see Questions below) with the discounting factor for rewards essentially acting as a soft-sliding-window to forget the past and Assumption 4.1 ensuring sufficient exploration to discover the changed rewards. Further, this setting can be viewed as a two timescale situation by considering the rewards that are sufficiently slow changing to be on slower timescale and the policy learning to be on the faster timescale thereby rendering convergence results here similar to convergence results in two timescale algorithms in stochastic optimization.

**Questions:**

- Non-Stationary Reinforcement Learning literature [1,2,3] (where the rewards and the transition probabilities are changing) characterize the static/dynamic regret. Could the authors comment further on the choice of the rate of convergence as the metric of analysis?
- Although in the slightly different setting of average reward, two-timescale, natural actor-critic, [4] analyzes the dynamic regret under evolving rewards and transition probabilities with no limitation on how they change
- Techniques to handle changing rewards aided by smoothness and boundedness have been studied in adjacent literature such as Performative RL [5] and adversarial RL [6, 7]. How do methods developed in this work compare with these?

References: \
[1] Cheung, W. C., Simchi-Levi, D., and Zhu, R. Reinforcement learning for non-stationary markov decision processes: The blessing of (more) optimism. ICML 2020. \
[2] Feng, S., Yin, M., Huang, R., Wang, Y.-X., Yang, J., and Liang, Y. Non-stationary reinforcement learning under general function approximation. ICML 2023. \
[3] Mao, W., Zhang, K., Zhu, R., Simchi-Levi, D., and Basar, T. Model-free nonstationary reinforcement learning: Near-optimal regret and applications in multiagent reinforcement learning and inventory control. Management Science 2024. \
[4] Jali, N., Pathak, E., Sharma, S., Qu, G., Joshi, G. Natural Policy Gradient for Average Reward Non-Stationary RL; 2025.
[5] Mandal, D., Triantafyllou, S., and Radanovic, G. Performative reinforcement learning. ICML 2023. \
[6] Chi Jin, Tiancheng Jin, Haipeng Luo, Suvrit Sra, and Tiancheng Yu. Learning adversarial markov decision processes with bandit feedback and unknown transition. ICML 2020. \
[7] Gergely Neu, Andras Antos, András György, and Csaba Szepesvári. Online markov decision processes under bandit feedback. NeurIPS 2010.

---

> ### Author Response · Authors · 2025-11-20
> **Response to Reviewer ziai (1/2)**
>
> Thanks for your time and effort in reviewing our paper! Please find our responses to your comments below. We will be happy to answer any further questions you may have.
>
> ### **Comparison with non-stationary RL and adversarial RL**
>
> - **Background and Motivation:** Non-stationary RL and adversarial RL are suited to online learning scenarios with non-stationary environments such as advertisement auctions, traffic management, etc. However, in contrast to the **online learning** regime, another class of real-world RL applications, including LLM fine-tuning and robotics, follow the **training-then-executing** regime. In these scenarios, various reward evolution techniques, including automated reward shaping, adaptive regularization, and curriculum learning, are widely used to train RL agents in order to address issues like under-exploration and credit assignment, and hence enhance training.
> - **Research Goal:** Non-stationary RL and adversarial RL focus more on the **algorithmic design** of an online learning problem to obtain a performance guarantee in adversarial environments. Our reward-evolving framework, however, focuses more on ensuring the convergence of existing practical RL algorithms by **providing sufficient conditions on the reward evolution** during the training phase.
> - **Choice of metric:** Non-stationary RL and adversarial RL characterize the static/dynamic regret as the performance of an algorithm during the online training process. In contrast, for the training-then-executing regime, we care more about the final convergence of the training phase and the performance in the executing phase. Consequently, we utilize the average squared gradient norm over the latter half of the training steps to characterize the convergence of the algorithm. Furthermore, as shown by previous literature [8,9], the stationarity of the policy often implies its optimality when the policy parameterization is specified. Thus, our chosen metric can also serve as a reasonable indicator of the optimality gap in the executing phase.
> - **Theoretical Results:** Given the above distinctions, it's hard to make a fair comparison on the theoretical results even for the dynamic-regret settings. This is because the forgetting mechanism,  which is their core algorithmic design, is unreasonable for the scenarios we consider. Without a forgetting mechanism, the algorithms proposed by [1,3,4] fail to achieve sub-linear regret. Additionally, these algorithms are limited to finite state-action spaces, while our analysis accommodates continuous spaces. The algorithm presented in [2] achieves a regret of $\tilde{O}(\sqrt{T})$ when assuming no forgetting, but this algorithm is impractical due to its dependence on updates and argmax operations upon the complex confidence set.
>
> ### **Connection and Comparison with Performative RL**
>
> Performative reward is a special case of our analysis. Since the reward (determined by occuancy measure, or equivalently the discounted visitation distribution) is assumed to be Lipschitz (Assumption 1 of [5]), we can treat a copy of the policy parameter $\theta$ as the reward parameter $\varphi$. This satisfies the conditions of our Corollary 4.8 (bounded gradient and $t^{-1/2}$ step sizes), resulting in $O(1/\sqrt{T})$ convergence, which aligns with the $\tilde{O}(\delta^{-4})$ sample complexity given by Theorem 3 of [5].  However, their approach requires exact solutions for a saddle point at each step, which is not feasible in practice.
>
> ### **Extensions of Our Results to Evolving Transitions**
>
> Since it is relatively uncommon for transition probabilities to be designed to evolve during the training phase in practice, we have not included this in our analysis. However, we believe that our approach can be adapted to accommodate evolving transitions, as long as we can establish the Lipschitz continuity for both $J(\cdot)$ and $\omega^*$ concerning the "transition parameter". We intend to explore this in future work.

---

> ### Author Response · Authors · 2025-11-20
> **Response to Reviewer ziai (2/2)**
>
> ### **Responses to Other Weaknesses and Questions**
>
> - W1: We appreciate your critical feedback and have included comparisons and connections in Section 2.
> - W2: We would like to clarify that our evolving-reward framework is a general model that accommodates both pre-defined reward scheduling and feedback-driven changes. Our convergence result hinges on the reward variation $F_T=\frac{1}{T/2}\sum_{t=T/2}^{T-1}||\varphi_{t+1}-\varphi_t||_2^2$. However, the specific manner in which $\varphi$ is updated can be general and is not detailed in our framework. It is also possible for $\varphi_t$ to change only at certain specific time steps, representing phase changes. As long as $F_T$ is bounded by a function of $T$, we can obtain a reasonable error bound.
> - W3 & Q3: We would like to clarify some misunderstandings:
>     - "the discounting factor for rewards essentially acting as a soft-sliding-window to forget the past": Our algorithm does not incorporate a forgetting mechanism. The changes in reward stem from algorithmic design intended to enhance training, so it is inappropriate to consider forgetting past experiences.
>     - "this setting can be viewed as a two timescale situation": Our framework also includes single-timescale scenarios that account for $\theta$, $\omega$, and $\varphi$. As stated in Corollary 4.8, when $\varphi$ employs a gradient-based update rule with bounded gradients and **step sizes proportional to the actor’s**, the algorithm can maintain $O(1/\sqrt{T})$ convergence as in the case with static rewards. (This misundersanding may primarily stem from a typo regarding the step sizes of the reward parameter in Corollary 4.8. It should be $\eta ^ \varphi _ t=\frac{c _ \varphi}{\sqrt{t}}$ instead of $\eta^ \varphi _ t=\frac{c _ \varphi}{t}$.)
>     - "the proof techniques have limited theoretical challenge (see Questions below)": To solely handle the changing rewards itself is trivial aided by smoothness and boundedness. The challenge lies in fitting the analysis of changing rewards into distinct algorithmic backbones and their analysis frameworks. As each of [5,6,7] built its algorithm and analysis upon a completely different algorithmic backbone from ours, controlling the impact of an evolving reward on the single-timescale actor-critic is non-trivial.
>
> We hope our responses fully address your concerns. If so, we wonder if you could kindly consider raising your rating score? We will also be happy to answer any further questions you may have. Thank you very much!
>
> ### References
> [8] Alekh Agarwal, Sham M. Kakade, Jason D. Lee, and Gaurav Mahajan. On the theory of policy gradient methods: optimality, approximation, and distribution shift. 2021.
>
> [9] Jincheng Mei, Chenjun Xiao, Csaba Szepesvári, and Dale Schuurmans. On the global convergence rates of softmax policy gradient methods. ICML 2020.

---

> ### Comment · Reviewer_ziai · 2025-11-21
>
> I thank the authors for their response and appreciate the expanded Section 2, justification of the motivation of the training-then-execution framework and the clarification of convergence as the performance metric. I would like to keep my score.

---

> > ### Author Response · Authors · 2025-11-24
> >
> > We sincerely appreciate your constructive suggestions for improving this paper. We believe we have addressed the concerns you raised. Do you have any remaining questions or concerns? We would be happy to address them. If not, could you please consider adjusting your rating score? Thank you once again for your time!

---

### Official Review · Reviewer_XQ6u · 2025-10-29

**Soundness:** 2
**Presentation:** 2
**Contribution:** 1
**Rating:** 2
**Confidence:** 4

**Summary:**

This paper provides the finite-time analysis of a single-timescale actor-critic with evolving reward.

**Strengths:**

-It is the first work considering single-timescale actor-critic with an evolving reward function.

**Weaknesses:**

This paper raises several significant concerns.

1. Assumption 4.1 is questionable. The definition of $A_\theta$ differs from all cited references, where the expectation is taken over the stationary distribution, whereas the authors instead take the expectation over the discounted state-visitation distribution. This formulation is highly uncommon and requires a strong, well-justified explanation.

 2. The claimed improvement over previous results by a factor of $\log^2 T$ is incomprehensible. The authors attribute this improvement to Proposition 4.8, which they state is independent of the mixing time of the ergodic Markov chain. However, this is not the source of the $\log^2 T$ term in prior works (see Wu et al., 2020; Chen \& Zhao, 2023; Tian et al., 2023; Chen \& Zhao, 2025). The authors should therefore provide a direct comparison with prior analyses and clarify how exactly they eliminate the $\log^2 T$ error term, which was indispensable in earlier studies.

 3. The contribution of this work appears incremental. The only notable difference from previous studies is the introduction of an evolving reward. However, no novel technical tools have been developed to address the evolving-reward setting, as its properties are already guaranteed by the exhaustive Assumption 4.5.

**Questions:**

See weaknesses.

---

> ### Author Response · Authors · 2025-11-20
> **Response to Reviewer XQ6u**
>
> Thanks for your time and effort in reviewing our paper! Please find our responses to your comments below. We will be happy to answer any further questions you may have.
> ### **On Assumption 4.1**
>
> In our revision, we have revised our assumption to state that
> $\Sigma_\theta=\mathbb{E} _ {\nu ^ {\pi _ \theta} _ \rho}[\phi(s)\phi(s)^\top]$
> is $\lambda$-strongly positive definite.
> This statement provides a more natural mathematical interpretation of "exploration" or "state-coverage" of the discounted state-visitation distribution.
> From this assumption, we can derive that the minimal eigenvalue of $A_\theta$ is lower-bounded by $(1-\sqrt{\gamma})\lambda$.
> A rigorous proof can be found in Appendix D.
>
> ### **On the $\log^2T$ factor**
>
> We apologize for any confusion caused. Previous work utilized the ergodicity of the Markov chain to control the Markovian noise. In contrast, we directly employ the contractivity of the transition mapping, which is a stronger condition than ergodicity. We have revised the relevant sections for better clarity.
>
> Below, we briefly compare the differences in the analysis:
>
> In prior research, the authors often needed to control $||\hat\mu_t-\mu_t||$.
> Given that they relied only on ergodicity, they needed to trace back $\tau$ steps for error decomposition into three components: $||\hat\mu_t-\mathcal{P} _ {t-\tau} ^\tau\hat\mu_t||$, $||\mathcal{P} _ {t-\tau} ^ \tau \hat\mu_t-\mu _ {t-\tau}||$, and $||\mu_{t-\tau}-\mu_t||$, where $\tau=\Theta(\log T)$ is the mixing time.
> The $\log^2T$ factor originates from the first component of this decomposition.
>
> In our work, we instead control $||\hat{\nu} _ t - \nu_t||$.
> Thanks to the contraction property, we only need to look back one step, breaking down the error into $||\hat{\nu} _ t - \nu_{t-1}||$ and $||\nu_{t-1} - \nu_t||$.
> The first term equals $||\hat{\mathcal{P}} _ {t-1} \hat\nu _ {t-1} - \hat{\mathcal{P}} _ {t-1} \nu _ {t-1}|| \leq \gamma || \hat \nu _ {t-1} - \nu _ {t-1}||$, thus eliminating the $\log^2T$ error term.
>
> ### **On the contributions of this work (W3)**
>
> We would like to emphasize that our contributions are threefold:
>
> Firstly, as reward evolution has been a very common case in practice yet largely overlooked in theoretical analysis, to introduce an evolving reward into the theoretical framework is an essential step towards bridging the theoretical-empirical gap in RL.
>
> Secondly, the upper bounds  on the shifts in reward parameters to ensure convergence or preserve the convergence rate, provide theoretical support to numerous practical RL techniques and serve as useful criteria for designing evolving rewards.
>
> Thirdly, our analysis of the Markovian noise is novel and will likely serve as a foundation for future studies in this area.
>
>
> We hope our response addresses your concerns. If so, we wonder if you could kindly consider raising your score? We will also be happy to answer any further questions you may have. Thank you very much!

---

### Official Review · Reviewer_SWb4 · 2025-10-30

**Soundness:** 3
**Presentation:** 3
**Contribution:** 3
**Rating:** 6
**Confidence:** 4

**Summary:**

This paper presents the first finite-time convergence analysis of a single-timescale actor-critic algorithm in a setting where the reward function evolves over time. This non-stationary reward setting is relevant to practical RL techniques like reward shaping, entropy regularization, and curriculum learning. The authors analyze a scenario where reward parameters can change at each time step, affecting both policy optimization (actor) and value estimation (critic). Under standard assumptions (linear function approximation for the critic, Lipschitz continuity of policy and reward, sufficient exploration), they derive non-asymptotic bounds for both actor and critic errors. The main result shows that an O(1/√T) convergence rate is achievable, matching the best-known rate for static rewards, provided the total variation of the reward parameters (FT) is sufficiently small (specifically, O(1/T) to preserve the rate).

**Strengths:**

1. The paper formalizes an important yet under-theorized problem. Providing theoretical guarantees for RL algorithms with evolving rewards directly addresses the gap between empirical practice and theoretical foundations.
2. This is the first work to provide finite-time convergence guarantees for actor-critic methods in the presence of evolving rewards under challenging Markovian sampling. The analysis is non-trivial and represents a substantial step forward in the theory of RL.
3. The novel analysis of distribution mismatch (Proposition 4.8) that improves the bound for the static-reward case is a meaningful secondary contribution, demonstrating the paper's technical depth beyond its primary focus.

**Weaknesses:**

1. Strong Assumptions:​​ The analysis relies on several strong assumptions that may limit its applicability to very complex environments. The most significant is the ​​linear function approximation​​ for the critic, which is a common but restrictive starting point. The theoretical community is increasingly focused on non-linear (neural network) approximations. The Lipschitz continuity assumptions, while standard, can also be difficult to verify or enforce in practice.
2. ​​Narrow Scope of Reward Evolution:​​ The analysis requires the reward to evolve "slowly enough." While this is a natural necessary condition, it may not capture all interesting practical scenarios where rewards might change in a more abrupt, phase-based manner (e.g., in some curriculum learning setups).

**Questions:**

What is the convergence result for AC with neural network approximation, as it has been studied by previous theoretical work?

---

> ### Author Response · Authors · 2025-11-20
> **Response to Reviewer SWb4**
>
> Thanks for your time and effort in reviewing our paper! Please find our responses to your comments below. We will be happy to answer any further questions you may have.
> - **On Our Assumptions (W1):**
> We are aware of recent works on TD learning[1] and actor-critic[2,3] that utilize deep neural networks for value function approximation, achieving convergence rates of $\tilde{O}(m^{-0.5})$ with respect to network width $m$. However, extending this analysis to single-timescale actor-critic[3] necessitates additional assumptions to ensure the tractability of the optimal critic parameter, which is beyond the primary focus of our work. We believe that extending our analysis to encompass a broader range of function approximators under more relaxed assumptions will be an interesting avenue for future research.
> - **On Reward Evolution (W2):**
> We would like to clarify that our evolving-reward framework is a general model that accommodates both slowly evolving rewards and abruptly changing, phase-based rewards. Our convergence result hinges on the reward variation $F_T=\frac{1}{T/2}\sum_{t=T/2}^{T-1}||\varphi_{t+1}-\varphi_t||_2^2$. It is possible for $\varphi_t$ to change only at a few specific time steps (representing a phase change). As long as $F_T$ is bounded by a function of $T$, we can obtain a reasonable error bound.
>
> We hope our responses fully address your concerns. If so, we wonder if you could kindly consider raising your rating score? We will also be happy to answer any further questions you may have. Thank you very much!
>
> ### References
>
> [1] Qi Cai, Zhuoran Yang, Jason D. Lee, and Zhaoran Wang. Neural temporal-difference learning converges to global optima. NeurIPS 2019.
>
> [2] Mudit Gaur, Amrit Singh Bedi, Di Wang, and Vaneet Aggarwal. Closing the gap: achieving global convergence (last iterate) of actor-critic under markovian sampling with neural network parametrization. ICML 2024.
>
> [3] Haoxing Tian, Ioannis Ch. Paschalidis, and Alex Olshevsky. Convergence of actor-critic methods with multi-layer neural networks. NeurIPS 2023.

---

### Official Review · Reviewer_T5XW · 2025-10-31

**Soundness:** 2
**Presentation:** 2
**Contribution:** 2
**Rating:** 4
**Confidence:** 3

**Summary:**

This work proposes an actor–critic algorithm with an evolving reward function and provides a finite-time convergence analysis. By introducing a KL-based regularization term, the authors isolate the dynamic component into a new parameter, denoted as $\varphi$, and derive theoretical results characterizing the algorithm’s performance as $\varphi$ evolves over time.

**Strengths:**

1. This work addresses the problem by introducing new parameters to capture the changing components, and derives results that characterize the system’s behavior through the evolution of these parameters.

2. The work further provides theoretical guarantees supporting the proposed approach.

**Weaknesses:**

The novelty of this work appears questionable, and the final results do not align with the stated goals. From the formulation section, the problem is presented as a standard Markov Decision Process (MDP) with a KL-regularized evolving reward function. In this formulation, the evolution of the reward arises solely from the regularization term. If the regularizer is deterministic, the optimal policy should be uniquely determined. However, in the algorithmic and theoretical sections, the authors introduce a new parameter, denoted as $\varphi$, to absorb the evolution of the reward function and update $\varphi$ in an arbitrary manner. This treatment deviates from the original problem formulation, leading to an inconsistency between the theoretical analysis and the stated objectives.

**Questions:**

1. Why is the parameter $\varphi$ allowed to change arbitrarily? What is the theoretical justification or practical motivation for treating $\varphi$ as a freely evolving variable?

2. Is the evolving reward solely induced by the regularization term, or is the reward function itself assumed to be adversarial or non-stationary? Clarifying this distinction is essential to understand the problem setting and the validity of the proposed analysis.

---

> ### Author Response · Authors · 2025-11-20
> **Response to Reviewer T5XW (1/2)**
>
> Thank you for your time and effort in reviewing our paper! We are grateful for your constructive suggestions, which have significantly guided our improvements. Please find our responses to your comments below.
>
> We feel sorry for causing the confusion regarding the fundamental setting of this paper——**the reward function itself is assumed to be non-stationary, but not adversarial**.
>
> As stated in Section 3.3, a regularized reward can be parameterized as $$\tilde r_{\varphi,\theta}(s,a)=r(s,a;\varphi)-\alpha(\varphi)\log\pi_\theta(a|s).$$
> Its evolution arises from three components: (i) the base reward $r(s,a)$, (ii) the regularization coefficient $\alpha$, and (iii) the current policy $\pi_\theta$. The parameters  $\varphi$   represent the first two components, as they share a similar theoretical analysis.
>
> Our formulation is motivated by three popular techniques in practical RL training (Section 1, Lines 45-53):
>
> - Automated reward shaping: corresponds to changes in $r(s,a;\varphi)$
> - Adaptive regularization: corresponds to changes in $\alpha(\varphi)$
> - Curriculum learning: corresponds to changes in $r(s,a;\varphi)$
>
> To facilitate the analysis of various reward evolution strategies—including pre-defined reward scheduling and feedback-driven changes—we assume that the reward can change arbitrarily and as frequently as other parameters. Consequently, our main theorem applies to all these scenarios (see Section 4.2).
>
> Notably, in the standard regularized Markov Decision Process (MDP) setting, although the regularized reward $\tilde{r}(s,a)$  evolves along with $\theta$, the parameters $\varphi$ remain static, leading to $F_T \equiv 0$. Our main theorem thus recovers the $O(1\sqrt{T})$ convergence rate for a single-timescale actor-critic (AC) algorithm.

---

> ### Author Response · Authors · 2025-11-20
> **Response to Reviewer T5XW (2/2)**
>
> ### Comparison with non-stationary RL and adversarial RL
>
> Although our formulation may seem like non-stationary RL and adversarial RL at first glance, they are acturally different in the following aspects:
>
> - **Background and Motivation:** Non-stationary RL and adversarial RL are suited to online learning scenarios with non-stationary environments such as advertisement auctions, traffic management, etc. However, in contrast to the **online learning** regime, another class of real-world RL applications, including LLM fine-tuning and robotics, follow the **training-then-executing** regime. In these scenarios, various reward evolution techniques, including automated reward shaping, adaptive regularization, and curriculum learning, are widely used to train RL agents in order to address issues like under-exploration and credit assignment, and hence enhance training.
> - **Research Goal:** Non-stationary RL and adversarial RL focus more on the **algorithmic design** of an online learning problem to obtain a performance guarantee in adversarial environments. Our reward-evolving framework, however, focuses more on ensuring the convergence of existing practical RL algorithms by **providing sufficient conditions on the reward evolution** during the training phase.
> - **Choice of metric:** Non-stationary RL and adversarial RL characterize the static/dynamic regret as the performance of an algorithm during the online training process. In contrast, for the training-then-executing regime, we care more about the final convergence of the training phase and the performance in the executing phase. Consequently, we utilize the average squared gradient norm over the latter half of the training steps to characterize the convergence of the algorithm. Furthermore, as shown by previous literature [1,2], the stationarity of the policy often implies its optimality when the policy parameterization is specified. Thus, our chosen metric can also serve as a reasonable indicator of the optimality gap in the executing phase.
> - **Theoretical Results:** Given the above distinctions, it's hard to make a fair comparison on the theoretical results even for the dynamic-regret settings. This is because the forgetting mechanism,  which is their core algorithmic design, is unreasonable for the scenarios we consider. Without a forgetting mechanism, the algorithms proposed by [3,4,5] fail to achieve sub-linear regret. Additionally, these algorithms are limited to finite state-action spaces, while our analysis accommodates continuous spaces. The algorithm presented in [6] achieves a regret of $\tilde{O}(\sqrt{T})$ when assuming no forgetting, but this algorithm is impractical due to its dependence on updates and argmax operations upon the complex confidence set.
>
> Following your comments, we have revised the relevant paragraphs in our latest update to improve clarity.
>
> We hope our response addresses your concerns. If so, we wonder if you could kindly consider raising your score? We will also be happy to answer any further questions you may have. Thank you very much!
>
> ### References
>
> [1] Alekh Agarwal, Sham M. Kakade, Jason D. Lee, and Gaurav Mahajan. On the theory of policy gradient methods: optimality, approximation, and distribution shift. 2021.
>
> [2] Jincheng Mei, Chenjun Xiao, Csaba Szepesvári, and Dale Schuurmans. On the global convergence rates of softmax policy gradient methods. ICML 2020.
>
> [3] Cheung, W. C., Simchi-Levi, D., and Zhu, R. Reinforcement learning for non-stationary markov decision processes: The blessing of (more) optimism. ICML 2020.
>
> [4] Mao, W., Zhang, K., Zhu, R., Simchi-Levi, D., and Basar, T. Model-free nonstationary reinforcement learning: Near-optimal regret and applications in multiagent reinforcement learning and inventory control. Management Science 2024.
>
> [5] Jali, N., Pathak, E., Sharma, S., Qu, G., Joshi, G. Natural Policy Gradient for Average Reward Non-Stationary RL; 2025.
>
> [6] Feng, S., Yin, M., Huang, R., Wang, Y.-X., Yang, J., and Liang, Y. Non-stationary reinforcement learning under general function approximation. ICML 2023.

---

### Author Response · Authors · 2025-11-26

Dear Reviewer,

We greatly appreciate the effort and expertise you have contributed to reviewing our paper. Since the author reviewer discussion period is ending soon, we hope that our responses have addressed your concerns. If that is the case, we wonder if you could kindly consider raising your score rating? Should you have any additional questions, we are more than happy to provide further assistance.

Thank you very much for the support!

---

### Meta-Review · Area_Chair_eK8U · 2026-01-06

**Summary:**

The paper studies the finite-time convergence of a single-timescale actor-critic algorithm in environments with evolving reward functions. The authors derive non-asymptotic bounds for actor and critic errors under Markovian sampling, showing that an $O(1/\sqrt{T})$ rate is achievable if the reward evolves slowly.

**Reviewer Concerns:**

While the theoretical rigor was generally acknowledged, most reviewers expressed significant concerns regarding the novelty of the technical tools and the restrictiveness of the assumptions. Specifically, the reviewers felt the contribution was incremental, as the evolving-reward properties appeared largely guaranteed by exhaustive assumptions rather than new analytical techniques.

**Reviewer Scores:**

Reviewer SWb4: 6 (Marginally Accept). Likely to stay at 6; appreciated the formalization of the problem but noted strong assumptions.

Reviewer ziai: 4 (Marginally Below). Stated they would keep their score after the rebuttal, citing limited theoretical challenge in the proof techniques.

Reviewer T5XW: 4 (Marginally Below). Initially concerned about the consistency between the formulation and algorithmic treatment; rebuttal provided some clarity.

Reviewer XQ6u: 2 (Reject). Likely to stay at 2; viewed the contribution as incremental and questioned the uniqueness of the analytical tools.

---

### Decision · Program_Chairs · 2026-01-26

Reject